# CLASS2SIMI: A NEW PERSPECTIVE ON LEARNING WITH LABEL NOISE

## ABSTRACT

Label noise is ubiquitous in the era of big data. Deep learning algorithms can easily fit the noise and thus cannot generalize well without properly modeling the noise. In this paper, we propose a new perspective on dealing with label noise called "*Class2Simi*". Specifically, we transform the training examples with noisy class labels into pairs of examples with noisy similarity labels, and propose a deep learning framework to learn robust classifiers with the noisy similarity labels. Note that a class label shows the class that an instance belongs to; while a similarity label indicates whether or not two instances belong to the same class. It is worthwhile to perform the transformation: We prove that the noise rate for the noisy similarity labels is lower than that of the noisy class labels, because similarity labels themselves are robust to noise. For example, given two instances, even if both of their class labels are incorrect, their similarity label could be correct. Due to the lower noise rate, Class2Simi achieves remarkably better classification accuracy than its baselines that directly deals with the noisy class labels.

## 1 INTRODUCTION

It is expensive to label large-scale data accurately. Therefore, cheap datasets with label noise are ubiquitous in the era of big data. However, label noise will degenerate the performance of trained deep models, because deep networks will easily overfit label noise (Zhang et al., 2017; Zhong et al., 2019; Li et al., 2019; Yi & Wu, 2019; Zhang et al., 2019; 2018; Xia et al., 2019; 2020).

In this paper, we propose a new perspective on handling label noise called "*Class2Simi*", i.e., transforming training examples with noisy class labels into pairs of examples with noisy similarity labels. A class label shows the class that an instance belongs to, while a similarity label indicates whether or not two instances belong to the same class. This transformation is motivated by the observation that the noise rate becomes lower, e.g., even if two instances have incorrect class labels, their similarity label could be correct. In the label-noise learning community, a lower noise rate usually results in higher classification performance (Han et al., 2018b; Patrini et al., 2017).

Specifically, we illustrate the transformation and the robustness of similarity labels in Figure 1. Assume we have eight noisy examples $\{(x_1, \bar{y}_1), \ldots, (x_8, \bar{y}_8)\}$ as shown in the upper part of the middle column. Their labels are of four classes, i.e., $\{1, 2, 3, 4\}$. The labels marked in red are incorrect labels. We transform the 8 examples into $8 \times 8$ example-pairs with noisy similarity labels as shown in the bottom part of the middle column, where the similarity label 1 means the two instances have the same class label and 0 means the two instances have different class labels. We present the latent clean class labels and similarity labels in the left column. In the middle column, we can see that although the instances $x_2$ and $x_4$ both have incorrect class labels, the similarity label of the example-pair $(x_2, x_4)$ is correct. Similarity labels are robust because they further consider the information on the pairwise relationship. We prove that the noise rate in the noisy similarity labels is lower than that of the noisy class labels. For example, if we assume that the noisy class labels in Figure 1 are generated according to the latent clean labels and the transition matrix shown in the upper part of the right column (the $ij$-th entry of the matrix denotes the probability that the clean class label $i$ flips into the noisy class label $j$), the noise rate for the noisy class labels is $0.5$ while the rate for the corresponding noisy similarity labels is $0.25$. Note that the noise rate is the ratio of the number of incorrect labels to the number of total examples, which can be calculated from the noise transition matrix combined with the proportion of each class, i.e., $1/6 \times 3/4 + 1/2 \times 1/4 = 0.25$.

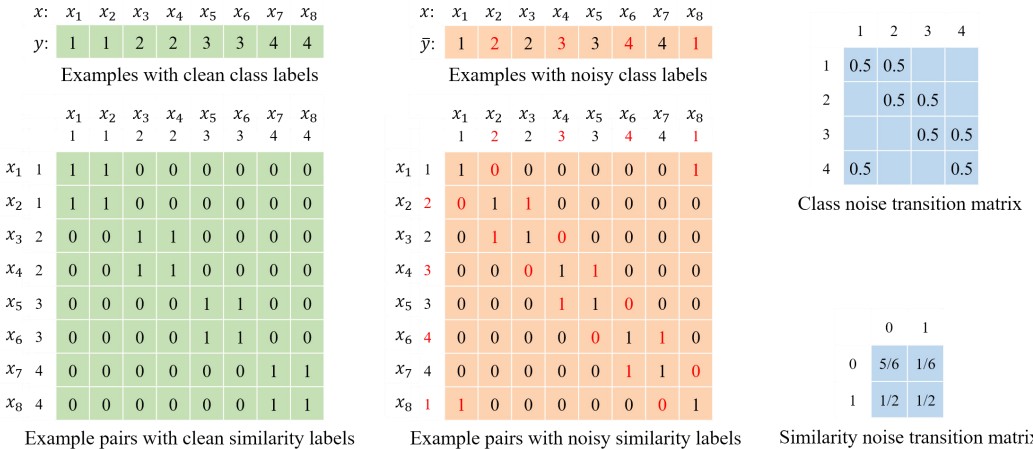

Figure 1: Illustration of the transformation from class labels to similarity labels. Note that $\bar{y}$ stands for the noisy class label and $y$ for the latent clean class label. The labels marked in red are incorrect labels. If we assume the class label noise is generated according to the noise transition matrix presented in the upper part of the right column, it can be calculated that the noise rate for the noisy class labels is 0.5 while the rate for the noisy similarity labels is 0.25. Note that the noise transition matrix for similarity labels can be calculated by exploiting the class noise transition matrix as in Theorem 1.

It is obvious that Class2Simi suffers information loss because we can not recover the class labels from similarity labels. However, since the similarity labels are more robust to noise than the class labels, the advantage of the reduction of noise rate overweighs the disadvantage of the loss of information. Intuitively, in the learning process, it is the signal in the information that enhances the performance of the model, while the noise in the information is harmful to the model. Through Class2Simi, although the total amount of information is reduced, the signal to noise ratio is increased, and so would be the total amount of signals. Thus, we can benefit from the transformation and achieve better performances. Theorem 2 and the experimental results will verify the effectiveness of this transformation.

It remains unsolved how to learn a robust classifier from the data with transformed noisy similarity labels. To solve this problem, we first estimate the similarity noise transition matrix, a $2 \times 2$ matrix whose entries denote the flip rates of similarity labels. Note that the transition matrix bridges the noisy similarity posterior and the clean similarity posterior. The noisy similarity posterior can be learned from the data with noisy similarity labels. Then, given the similarity noise transition matrix, we can infer the clean similarity posterior from the noisy similarity posterior. Since the clean similarity posterior is approximated by the inner product of the clean class posterior (Hsu et al., 2019), the clean class posterior (and thus the robust classifier) can thereby be learned. We will empirically show that Class2Simi with the estimated similarity noise transition matrix will remarkably outperform the baselines even given with the ground-truth class noise transition matrix.

The contributions of this paper are summarized as follows:

- We propose a new perspective on learning with label noise, which transforms class labels into similarity labels. Such a transformation reduces the noise level.
- We provide a way to estimate the similarity noise transition matrix by theoretically establishing its relation to the class noise transition matrix. We show that even if the class noise transition matrix is inaccurately estimated, the induced similarity noise transition matrix still works well.
- We design a deep learning method to learn robust classifiers from data with noisy similarity labels and theoretically analyze its generalization ability.
- We empirically demonstrate that the proposed method remarkably surpasses the baselines on many datasets with both synthetic noise and real-world noise.

The rest of this paper is organized as follows. In Section 2, we formalize the noisy multi-class classification problem, and in Section 3, we propose the Class2Simi strategy and practical implementation. Experimental results are discussed in Section 4. We conclude our paper in Section 5.

## 2 PROBLEM SETUP AND RELATED WORK

Let $(X, Y) \in \mathcal{X} \times \{1, \ldots, C\}$ be the random variables for instances and clean labels, where $\mathcal{X}$ represents the instance space and $C$ is the number of classes. However, in many real-world applications (Zhang et al., 2017; Zhong et al., 2019; Li et al., 2019; Yi & Wu, 2019; Zhang et al., 2019; Tanno et al., 2019; Zhang et al., 2018), the clean labels cannot be observed. The observed labels are noisy. Let $\bar{Y}$ be the random variable for the noisy labels. What we have is a sample $\{(x_1, \bar{y}_1), \ldots, (x_n, \bar{y}_n)\}$ drawn from the noisy distribution $\mathcal{D}_\rho$ of the random variables $(X, \bar{Y})$. Our aim is to learn a robust classifier that could assign clean labels to test data by exploiting the sample with noisy labels.

Existing methods for learning with noisy labels can be divided into two categories: algorithms that result in statistically inconsistent or consistent classifiers. Methods in the first category usually employ heuristics to reduce the side-effect of noisy labels, e.g., selecting reliable examples (Yu et al., 2019; Han et al., 2018b; Malach & Shalev-Shwartz, 2017), reweighting examples (Ren et al., 2018; Jiang et al., 2018; Ma et al., 2018; Kremer et al., 2018; Tanaka et al., 2018; Reed et al., 2015), employing side information (Vahdat, 2017; Li et al., 2017; Berthon et al., 2020), and adding regularization (Han et al., 2018a; Guo et al., 2018; Veit et al., 2017; Vahdat, 2017; Li et al., 2017). Those methods empirically work well in many settings. Methods in the second category aim to learn robust classifiers that could converge to the optimal ones defined by using clean data. They utilize the noise transition matrix, which denotes the probabilities that the clean labels flip into noisy labels, to build consistent algorithms (Goldberger & Ben-Reuven, 2017; Patrini et al., 2017; Thekumparampil et al., 2018; Yu et al., 2018; Liu & Guo, 2020; Zhang & Sabuncu, 2018; Kremer et al., 2018; Liu & Tao, 2016; Northcutt et al., 2017; Scott, 2015; Natarajan et al., 2013; Yao et al., 2020b). The idea is that given the noisy class posterior probability and the noise transition matrix, the clean class posterior probability can be inferred.

Note that the noisy class posterior and the noise transition matrix can be estimated by exploiting the noisy data, where the noise transition matrix additionally needs anchor points (Liu & Tao, 2016; Patrini et al., 2017). Some methods assume anchor points have already been given (Yu et al., 2018). There are also methods showing how to identify anchor points from the noisy training data (Liu & Tao, 2016; Patrini et al., 2017).

## 3 CLASS2SIMI MEETS NOISY SUPERVISION

In this section, we propose a new strategy for learning from noisy data. Our core idea is to transform class labels to similarity labels first, and then handle the noise manifested on similarity labels.

### 3.1 TRANSFORMATION ON LABELS AND THE TRANSITION MATRIX

As in Figure 1, we combine every 2 instances in pairs, and if the two instances have the same class label, we assign this pair a similarity label 1, otherwise 0. If the class labels are corrupted, the generated similarity labels also contain noise. We denote the clean and noisy similarity labels of the example-pair $(x_i, x_j)$ by $H_{ij}$ and $\bar{H}_{ij}$ respectively.

The definition of the similarity noise transition matrix is similar to the class one, denoting the probabilities that clean similarity labels flip into noisy similarity labels, i.e., $T_{s,mn} = P(\bar{H}_{ij} = n | H_{ij} = m)$. The dimension of the similarity noise transition matrix is always $2 \times 2$. Since the similarity labels are generated from class labels, the similarity noise is also determined and, thus can be calculated, by the class noise transition matrix.

**Theorem 1.** *Assume that the dataset is balanced (each class has the same amount of samples, and c classes in total), and the noise is class-dependent. Given a class noise transition matrix $T_c$, such that $T_{c,ij} = P(\bar{Y} = j | Y = i)$. The elements of the corresponding similarity noise transition matrix $T_s$ can be calculated as*

$$T_{s,00} = \frac{c^2 - c - \left(\sum_j (\sum_i T_{c,ij})^2 - ||T_c||^2_{\text{Fro}}\right)}{c^2 - c}, \qquad T_{s,01} = \frac{\sum_j (\sum_i T_{c,ij})^2 - ||T_c||^2_{\text{Fro}}}{c^2 - c},$$

$$T_{s,10} = \frac{c - ||T_c||^2_{\text{Fro}}}{c}, \qquad T_{s,11} = \frac{||T_c||^2_{\text{Fro}}}{c}.$$

Figure 2: An overview of the proposed method. We add a pairwise enumeration layer and similarity transition matrix to calculate and correct the predicted similarity posterior. By minimizing the proposed loss $L_{c2s}$, a classifier $f$ can be learned for assigning clean labels. The detailed structures of the Neural Network are provided in Section 4. Note that for the noisy similarity labels, some of them are correct and some are not. The similarity label for dogs is correct and the similarity label for cats is incorrect. In practice, the input data is original class-labeled data, and the transformation is conducted during the training procedure rather than before training.

A detailed proof is provided in Appendix A.

**Remark 1.** *Theorem 1 can easily extend to the setting where the dataset is unbalanced in classes by multiplying each $T_{c,ij}$ by a coefficient $n_i$. $n_i$ is the number of examples from the $i$-th class.*

Note that the similarity labels are only dependent on class labels. If the class noise is class-dependent, the similarity noise is also "class-dependent" (class means similar and dissimilar). Under class-dependent label noise, a binary classification is learnable as long as $T_{00} + T_{11} > 1$ (Menon et al., 2015), where $T$ is the corresponding binary transition matrix; a multi-class classification is learnable if the corresponding transition matrix $T_c$ is invertible. For Class2Simi, in the most general sense, i.e., $T_c$ is invertible, $T_{s,00} + T_{s,11} > 1$ holds. Namely, the learnability of the pointwise classification implies the learnability of the reduced pairwise classification. However, the latter cannot imply the former. A proof and a counterexample are provided in Appendix F.

**Theorem 2.** *Assume that the dataset is balanced (each class has the same amount of samples), and the noise is class-dependent. When the number of classes $c \geq 8$[1], the noise rate for the noisy similarity labels is lower than that of the noisy class labels.*

A detailed proof is provided in Appendix B.

When dealing with label noise, a low noise rate has many benefits. The most important one is that the noise-robust algorithms will consistently achieve higher performance when the noise rate is lower (Bao et al., 2018; Han et al., 2018b; Xia et al., 2019; Patrini et al., 2017). Another benefit is that, when the noise rate is low, the complex instance-dependent label noise can be well approximated by class-dependent label noise (Cheng et al., 2020), which is easier to handle. After the Class2Simi transformation, the number of dissimilar pairs is $(c-1)$ times as much as that of similar pairs. Meanwhile, compared with the original noise rate of class labels, the noise rate of similar pairs (the ratio of the number of mislabeled similar pairs to the number of total real similar pairs) is higher and the noise rate of dissimilar pairs is lower, while the overall noise rate of pairwise examples is lower, which partially reflects that the impact of the label noise is less bad. Moreover, the flip from dissimilar to similar should be more adversarial and thus more important. In practice, it is common that one class has more than one clusters, while it is rare that two or more classes are in the same cluster. If there is a flip from similar to dissimilar and based on it we split a (latent) cluster into two (latent) clusters, we still have a high chance to label these two clusters correctly later. If there is a flip from dissimilar to similar and based on it we join two clusters belonging to two classes into a single cluster, we nearly have zero chance to label this cluster correctly later. As a consequence, the flip from dissimilar to similar is more adversarial, and thus more important. To sum up, considering the reduction of the overall noise rate is meaningful.

## 3.2 LEARNING WITH NOISY SIMILARITY LABELS

In order to learn a multi-class classifier from similarity labeled data, we should establish relationships between class posterior probability and similarity posterior probability. Here we employ the relationship established in (Hsu et al., 2019), which is derived from a likelihood model. As in Figure

---

[1] In multi-class classification problems, the number of classes is usually bigger than 8 , e.g., *MNIST* (LeCun, 1998), *CIFAR-10*, and *CIFAR-100* (Krizhevsky et al., 2009).

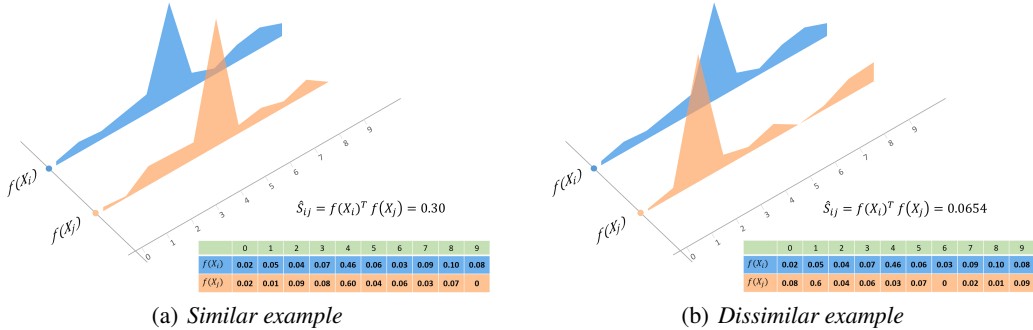

(a) *Similar example*  (b) *Dissimilar example*

Figure 3: Examples of predicted noisy similarity. Assume class number is 10; $f(X_i)$ and $f(X_j)$ are categorical distribution of $X_i$ and $X_j$ respectively, which are shown above in the form of area charts. $\hat{S}_{ij}$ is the predicted similarity posterior between two instances, calculated by the inner product between two categorical distributions.

2, they denote the predicted clean similarity posterior by the inner product between two categorical distributions: $\hat{S}_{ij} = f(X_i)^\top f(X_j)$. Intuitively, $f(X)$ outputs the predicted categorical distribution of input data $X$ and $f(X_i)^\top f(X_j)$ can measure how similar the two distributions are. For clarity, we visualize the predicted similarity posterior in Figure 3. If $X_i$ and $X_j$ are predicted belonging to the same class, i.e., $\mathrm{argmax}_{m \in C} f_m(X_i) = \mathrm{argmax}_{n \in C} f_n(X_j)$, the predicted similarity posterior should be relatively high ($\hat{S}_{ij} = 0.30$ in Figure 3(a)). By contrast, if $X_i$ and $X_j$ are predicted belonging to different classes, the predicted similarity posterior should be relatively low ($\hat{S}_{ij} = 0.0654$ in Figure 3(b)). Note that the noisy similarity posterior distribution $P(\bar{H}_{ij}|X_i, X_j)$ and clean similarity posterior distribution $P(H_{ij}|X_i, X_j)$ satisfy

$$P(\bar{H}_{ij}|X_i, X_j) = T_s^\top P(H_{ij}|X_i, X_j). \tag{1}$$

Therefore, we can infer noisy similarity posterior $\hat{\bar{S}}_{ij}$ from clean similarity posterior $\hat{S}_{ij}$ with the similarity noise transition matrix. To measure the error between the predicted noisy similarity posterior $\hat{\bar{S}}_{ij}$ and noisy similarity label $\bar{H}_{ij}$, we employ a binary cross-entropy loss function (Shannon, 1948). The final optimization function is

$$L_{c2s}(\bar{H}_{ij}, \hat{\bar{S}}_{ij}) = -\sum_{i,j} \bar{H}_{ij} \log \hat{\bar{S}}_{ij} + (1 - \bar{H}_{ij}) \log(1 - \hat{\bar{S}}_{ij}). \tag{2}$$

The pipeline of the proposed Class2Simi is summarized in Figure 2. The softmax function outputs an estimation for the clean class posterior, i.e., $f(X) = \hat{P}(Y|X)$, where $\hat{P}(Y|X)$ denotes the estimated class posterior. Then a pairwise enumeration layer (Hsu et al., 2018) is added to calculate the predicted clean similarity posterior $\hat{S}_{ij}$ of every two instances. According to Equation 1, by pre-multiplying the transpose of the noise similarity transition matrix, we can obtain the predicted noisy similarity posterior $\hat{\bar{S}}_{ij}$. Therefore, by minimizing $L_{c2s}$, we can learn a classifier for predicting noisy similarity labels. Meanwhile, before the transition matrix layer, the pairwise enumeration layer will output a prediction for the clean similarity posterior, which guides $f(X)$ to predict clean class labels.

### 3.3 IMPLEMENTATION

The proposed algorithm is summarized in Algorithm 1. Since learning only from similarity labels will lose the mapping between the output nodes and the semantic classes, we load the model trained on the data with noisy class labels to learn the class information in Stage 2. It is worthwhile to mention that Class2Simi increases the computation cost slightly. Note that the transformation of labels is during the training phase rather than before training. Specifically, as in Figure 2, first, we read a batch of $n$ examples, and generate their corresponding $n^2$ similarity labels. Since $n$ is the batch size, it is usually small. In addition, we only save the labels, not example-pairs, such that it introduces a

---

**Algorithm 1** Class2Simi

---

**Input**: training data with noisy class labels; validation data with noisy class labels.
**Stage 1: Learn $\hat{T}_s$**
1: Learn $g(X) = \hat{P}(\bar{Y}|X)$ by training data with noisy class labels, and save the model for Stage 2;
2: Estimate $\hat{T}_c$ following the optimization method in (Patrini et al., 2017);
3: Transform $\hat{T}_c$ to $\hat{T}_s$.
**Stage 2: Learn the classifier** $f(X) = \hat{P}(Y|X)$
4: Load the model saved in Stage 1, and train the whole pipeline showed in Figure 2.
**Output**: classifier $f$.

---

negligible memory overhead. Then the neural network outputs the class posterior probabilities of $n$ single examples in the batch of data. After that pairwise enumeration layer calculates the inner products between every two instances, outputting $n^2$ predicted similarity posterior probabilities. Then the similarity transition matrix corrects the $n^2$ predicted similarity posterior probabilities. Finally, the loss is accumulated by $n^2$ items. Namely, Class2Simi only does the additional computation on generating similarity labels and calculating the inner products between every two instances in the pairwise enumeration layer, which is time-efficient.

### 3.4 GENERALIZATION ERROR

We formulate the above problem in the traditional risk minimization framework (Mohri et al., 2018). The expected and empirical risks of employing estimator $f$ can be defined as

$$R(f) = E_{(X_i, X_j, \bar{Y}_i, \bar{Y}_j, \bar{H}_{ij}, T_s) \sim \mathcal{D}_\rho}[\ell(f(X_i), f(X_j), T_s, \bar{H}_{ij})], \qquad (3)$$

and

$$R_n(f) = \frac{1}{n^2} \sum_{i=1}^{n} \sum_{j=1}^{n} \ell(f(X_i), f(X_j), T_s, \bar{H}_{ij}), \qquad (4)$$

where $n$ is training sample size of the noisy data. Assume that the neural network has $d$ layers with parameter matrices $W_1, \ldots, W_d$, and the activation functions $\sigma_1, \ldots, \sigma_{d-1}$ are Lipschitz continuous, satisfying $\sigma_j(0) = 0$. We denote by $H : X \mapsto W_d \sigma_{d-1}(W_{d-1}\sigma_{d-2}(\ldots \sigma_1(W_1 X))) \in \mathcal{R}$ the standard form of the neural network. $H = \text{argmax}_{i \in \{1, \ldots, C\}} h_i$. Then the output of the softmax function is defined as $f_i(X) = \exp(h_i(X)) / \sum_{j=1}^{C} \exp(h_j(X)), i = 1, \ldots, C$. We can obtain the following generalization error bound as follow.

**Theorem 3.** *Assume the parameter matrices $W_1, \ldots, W_d$ have Frobenius norm at most $M_1, \ldots, M_d$, and the activation functions are 1-Lipschitz, positive-homogeneous, and applied element-wise (such as the ReLU). Assume the transition matrix is given, and the instances $X$ are upper bounded by $B$, i.e., $\|X\| \leq B$ for all $X$, and the loss function $\ell$ is upper bounded by $M^2$. Then, for any $\delta > 0$, with probability at least $1 - \delta$,*

$$R(\hat{f}) - R_n(\hat{f}) \leq \frac{(T_{s,11} - T_{s,01})2BC(\sqrt{2d\log 2} + 1)\Pi_{i=1}^{d} M_i}{T_{s,11}\sqrt{n}} + M\sqrt{\frac{\log 1/\delta}{2n}}. \qquad (5)$$

Notation and a detailed proof are provided in Appendix C.

Theorem 3 implies that if the training error is small and the training sample size is large, the expected risk $R(\hat{f})$ of the representations for noisy similarity posterior will be small. If the transition matrix is well estimated, the clean similarity posterior as well as the classifier for the clean class will also have a small risk according to Equation 1 and the Class2Simi relations. This theoretically justifies why the proposed method works well. In the experiment section, we will show that the transition matrices are well estimated and that the proposed method significantly outperforms the baselines.

In Class2Simi, a multi-class classification is reduced to a pairwise binary classification. For pairwise examples, if a surrogate loss is classification-calibrated, minimizing it leads to minimizing the zero-one loss on the pointwise random variables in the limit case. Otherwise, we cannot guarantee the

---

[2]The assumption holds because deep neural networks will always regulate the objective to be a finite value and thus the corresponding loss functions are of finite values.

Table 1: Means and Standard Deviations of Classification Accuracy over 5 trials on image datasets.

| *MNIST* Sym-Noise | 0.1 | 0.2 | 0.3 | 0.4 | 0.5 |
|---|---|---|---|---|---|
| Co-teaching | 98.40±0.07 | 98.22±0.10 | 97.56±0.04 | 97.30±0.15 | 96.12±0.53 |
| APL | 98.77±0.07 | 98.62±0.12 | 98.55±0.11 | 98.40±0.12 | 98.05±0.29 |
| S2E | 98.13±0.57 | 98.18±0.21 | 97.71±0.18 | 97.41±0.17 | 96.63±0.29 |
| Forward | 98.64±0.18 | 98.33±0.23 | 98.20±0.11 | 97.99±0.20 | 97.35±0.46 |
| Forward & Class2Simi | **98.84±0.09** | **98.74±0.17** | **98.56±0.15** | **98.44±0.07** | **98.25±0.26** |
| Reweight | 98.23±0.27 | 98.01±0.17 | 97.72±0.30 | 97.66±0.45 | 96.81±0.70 |
| Reweight & Class2Simi | **98.51±0.13** | **98.07±0.29** | **98.08±0.18** | **97.82±0.29** | **97.11±0.20** |
| Revision | 98.59±0.12 | 98.48±0.20 | 98.15±0.17 | 97.94±0.16 | 97.53±0.31 |
| Revision & Class2Simi | **98.62±0.17** | **98.57±0.24** | **98.19±0.21** | **97.99±0.19** | **97.73±0.27** |
| *CIFAR10* Sym-Noise | 0.1 | 0.2 | 0.3 | 0.4 | 0.5 |
| Co-teaching | 85.16±0.25 | 83.59±0.17 | 80.47±0.39 | 78.42±0.25 | 74.35±0.84 |
| APL | 83.81±0.45 | 82.2±0.56 | 80.49±0.82 | 77.80±1.62 | 73.25±2.45 |
| S2E | 60.01±0.89 | 58.53±1.49 | 55.07±4.35 | 52.07±3.33 | 50.10±3.61 |
| Forward | 87.01±0.41 | 85.75±0.37 | 83.72±0.33 | 81.28±0.34 | 78.10±0.72 |
| Forward & Class2Simi | **87.84±0.12** | **86.62±0.20** | **84.89±0.19** | **83.32±0.72** | **81.15±0.32** |
| Reweight | 86.80±0.36 | 85.08±0.33 | 83.03±0.63 | 80.35±0.41 | 76.61±0.81 |
| Reweight & Class2Simi | **87.23±0.39** | **85.43±0.65** | **83.18±0.57** | **80.67±0.62** | **77.36±0.60** |
| Revision | 87.09±0.36 | 85.68±0.26 | 83.88±0.49 | 81.41±0.47 | 77.96±0.44 |
| Revision & Class2Simi | **87.48±0.43** | **85.87±0.58** | **83.92±0.37** | **81.87±0.44** | **78.70±0.96** |
| *CIFAR100* Sym-Noise | 0.1 | 0.2 | 0.3 | 0.4 | 0.5 |
| Co-teaching | 52.39±0.47 | 49.83±0.42 | 46.31±0.72 | 42.05±0.80 | 35.21±1.01 |
| APL | 37.70±1.72 | 33.35±2.07 | 28.80±2.58 | 24.82±2.79 | 21.27±1.49 |
| S2E | 49.30±1.93 | 46.20±2.10 | 43.24±2.48 | 39.63±1.86 | 34.98±1.87 |
| Forward | 52.63±0.48 | 45.67±0.94 | 42.25±1.83 | 37.42±1.45 | 30.66±1.31 |
| Forward & Class2Simi | **55.56±0.55** | **52.85±0.82** | **49.44±0.70** | **45.52±0.52** | **39.86±0.38** |
| Reweight | 51.43±0.22 | 47.01±0.83 | 42.62±0.66 | 36.02±2.40 | 26.34±0.96 |
| Reweight & Class2Simi | **51.74±3.65** | **49.57±1.60** | **46.54±3.20** | **43.65±2.14** | **34.01±3.49** |
| Revision | 51.48±0.22 | 47.11±0.87 | 42.75±0.78 | 36.08±2.52 | 26.32±0.94 |
| Revision & Class2Simi | **53.30±1.81** | **50.18±0.83** | **47.51±1.71** | **44.20±1.70** | **35.36±2.86** |

worst-case learnability of learning pointwise labels from pairwise examples, but it cannot imply the average-case non-learnability either. Theoretically, Bao et al. (2020) proved that when the pairwise labels are all correct, for the special case $c = 2$, a good model for predicting similar/dissimilar pairs must also be a good model for predicting the original classes, under mild assumptions. In practice, it seems fine to use non-classification-calibrated losses. According to Tewari & Bartlett (2007), the multi-class margin loss (i.e., one-vs-rest loss) and the pairwise comparison loss (i.e., one-vs-one loss) are proved to be non-calibrated, but they are still the main multi-class losses in Mohri et al. (2018); Shalev-Shwartz & Ben-David (2014).

## 4 EXPERIMENTS

**Datasets.** We employ three widely used image datasets, i.e., *MNIST* (LeCun, 1998), *CIFAR-10*, and *CIFAR-100* (Krizhevsky et al., 2009), one text dataset *News20*, and one real-world noisy dataset *Clothing1M* (Xiao et al., 2015). *MNIST* has $28 \times 28$ grayscale images of 10 classes including 60,000 training images and 10,000 test images. *CIFAR-10* and *CIFAR-100* both have $32 \times 32 \times 3$ color

images including 50,000 training images and 10,000 test images. *CIFAR-10* has 10 classes while *CIFAR-100* has 100 classes. *News20* is a collection of approximately 20,000 newsgroup documents, partitioned nearly evenly across 20 different newsgroups. *Clothing1M* has 1M images with real-world noisy labels and additional 50k, 14k, 10k images with clean labels for training, validation and test, and we only use noisy training set in the training phase. Note that the similarity learning method of Class2Simi is based on *Cluster* because there is no class information. Intuitively, for a noisy class, if most instances in it belong to another specific class, we can hardly identify it. For example, assume that a class with noisy labels $\bar{i}$ contains $n_i$ instances with ground-truth labels $i$ and $n_j$ instances with ground-truth labels $j$. If $n_j$ is bigger than $n_i$, the model will cluster class $i$ into $j$. Unfortunately, in *Clothing1M*, most instances with label '5' belong to class '3' actually. Therefore, we merge the two classes, and denote the fixed dataset by *Clothing1M** which contains 13 classes. For all the datasets, we leave out 10% of the training examples as a validation set, which is for model selection.

**Noisy class labels generation.** For the three clean datasets, we artificially corrupt the class labels of training and validation sets according to the class noise transition matrix. Specifically, for each instance with clean label $i$, we replace its label by $j$ with a probability of $T_{c,ij}$. In this paper, we consider both symmetric and asymmetric noise settings which are defined in Appendix D.

**Baselines.** As mentioned before, Class2Simi is a strategy rather than a specific algorithm. In this paper, we employ three $T$-based methods, i.e., Forward correction (Patrini et al., 2017), Reweight (Liu & Tao, 2016), and $T$-revision (Xia et al., 2019), which all utilize a class-dependent transition matrix to model the noise, to implement our approach to show the effectiveness of Class2Simi. Besides, we externally conduct experiments on Co-teaching (Han et al., 2018b), which is a representative algorithm of selecting reliable examples for training; APL (Ma et al., 2020), which applies simple normalization on loss functions and makes them robust to noisy labels; S2E (Yao et al., 2020a), which properly controls the sample selection process so that deep networks can benefit from the memorization effect.

**Network structure and Optimizer.** For *MNIST*, we use LeNet (LeCun et al., 1998). For *CIFAR-10*, we use ResNet-32 with pre-activation (He et al., 2016b). For *CIFAR-100*, we use ResNet-56 with pre-activation (He et al., 2016b). For *News20*, we use GloVe (Pennington et al., 2014) to obtain vector representations for text, and employ a 3-layer MLP with the Softsign active function. For *Clothing1M**, we use pre-trained ResNet-50 (He et al., 2016a). We use the same optimization method as Forward correction to learn the noise transition matrix $\hat{T}_c$. In Stage 2, we use the Adam optimizer with initial learning rate 0.001. On *MNIST*, the batch size is 128 and the learning rate decays every 20 epochs by a factor of 0.1 with 60 epochs in total. On *CIFAR-10*, the batch size is also 128 and the learning rate decays every 40 epochs by a factor of 0.1 with 120 epochs in total. On *CIFAR-100*, the batch size is 1000 and the learning rate drops at epoch 80 and 160 by a factor of 0.1 with 200 epochs in total. On *News20*, the batch size is 128 and the learning rate decays every 10 epochs by a factor of 0.1 with 30 epochs in total. On *Clothing1M**, the batch size is 32 and the learning rate drops every 5 epochs by a factor of 0.1 with 10 epochs in total.

**Results on noisy image datasets.** The results in Table 1 and Figure 4 demonstrate that Class2Simi achieves distinguished classification accuracy and is robust against the estimation errors on transition matrix.

From Table 1, overall, we can see that after the transformation, better performances are achieved due to a lower noise rate and the similarity transition matrix being robust to noise. Specifically, On *MNIST*, as the noise rate increases from Sym-0.1 to Sym-0.5, Forward & Class2Simi maintains remarkable accuracy above 98.20% while the accuracy of Forward decreases steadily. On *CIFAR100*, there are obvious decreases in the accuracy of all methods and our method achieves the best results across all noise rate, i.e., at Sym-0.5, Class2Simi gives accuracy uplifts of about 9.0% compared with those $T$-based methods. Results under asymmetric noise are provided in Appendix E.3.

In Figure 4, we show that the similarity noise transition matrix is robust against estimation errors. To verify this, we add some random noise to the ground-truth $T_c$ through multiplying every element in class $T_c$ by a random variable $\alpha_{ij}$. We control the noise rate on the $T_c$ by sampling $\alpha_{ij}$ in different intervals, i.e., 0.1 noise means that $\alpha_{ij}$ is uniformly sampled from $\pm[1.1, 1.2]$. Then we normalize $T_c$ to make its row sums equal to 1. From Figure 4, we can see that the accuracy of Forward drops dramatically with the increase of the noise on $T_c$ on three datasets. Meanwhile, there is only a slight fluctuation of Forward & Class2Simi on *MNIST*

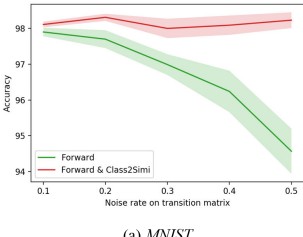 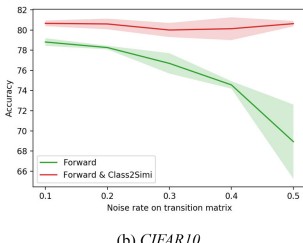 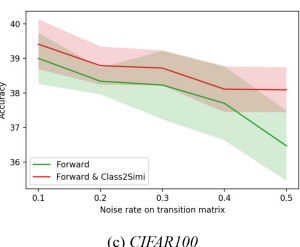

(a) *MNIST*  (b) *CIFAR10*  (c) *CIFAR100*

Figure 4: Means and Standard Deviations of Classification Accuracy over 5 trials on *MNIST*, *CIFAR10* and *CIFAR100* with perturbational ground-truth $\hat{T}_c$.

and *CIFAR10*. On *CIFAR100*, the accuracy of Forward & Class2Simi is affected by the noise on $T_c$, but the decline is much lower than Forward. The reason is that Forward & Class2Simi needs to learn the class information from noisy data which is hard when the number of classes is large.

**Results on noisy text dataset.** Results in Table 2 show that the proposed strategy works well on the text dataset under both symmetric and asymmetric noise settings.

**Results on real-world noisy dataset.** Results in Table 3 show that the proposed strategy significantly improves the classification accuracy of the $T$-based methods. $T$-based methods with Class2Simi also outperform those classic methods.

**Ablation study.** To investigate how the similarity loss function influences the classification accuracy, we conduct experiments with the cross-entropy loss function and the similarity loss function respectively on clean datasets over 3 trails where the $T_c$ is set to an identity matrix. All other settings are kept the same. As shown in Table 4, the similarity loss function does not improve the classification accuracy, which means the accuracy increase in our paper is benefited from the lower noise rate and the more robust transition matrix.

Table 2: Classification Accuracy on *News20*.

| *News20* | Sym-0.2 | Sym-0.4 | Asym-0.3 |
|---|---|---|---|
| Forward | 48.07±0.26 | 46.49±0.54 | 47.30±0.53 |
| F & C2S | **48.52±0.47** | **47.04±0.33** | **47.70±0.45** |
| Reweight | 48.30±0.44 | 46.34±0.31 | 47.25±0.91 |
| Rt & C2S | **48.55±0.46** | **47.71±0.58** | **48.43±0.61** |
| Revision | 48.25±0.43 | 46.32±0.19 | 47.40±0.76 |
| Rn & C2S | **48.63±0.48** | **47.84±0.64** | **48.53±0.53** |

*(CE uses class labels and the cross-entropy loss function.)*

Table 3: Classification Accuracy on *Clothing1M\**.

| CE | 72.49 | S2E | 72.30 |
|---|---|---|---|
| APL | 58.93 | Co-teaching | 74.70 |
| Forward | 73.88 | Forward & Class2Simi | 75.41 |
| Reweight | 74.44 | Reweight & Class2Simi | 75.76 |
| Revision | 74.65 | Revision & Class2Simi | **75.79** |

Table 4: Classification Accuracy on clean datasets.

| Dataset | MNIST | CIFAR10 | CIFAR100 | News20 |
|---|---|---|---|---|
| CE | **99.19±0.07** | 89.09±0.19 | 56.12±0.93 | **49.29±0.33** |
| C2S | 99.10±0.13 | **89.18±0.25** | **56.17±0.37** | 48.71±0.56 |

## 5 CONCLUSION

This paper proposes a new perspective on dealing with class label noise (called Class2Simi) by transforming the training sample with noisy class labels into a training sample with noisy similarity labels. We also propose a deep learning framework to learn classifiers directly with the noisy similarity labels. The core idea is to transform class information into similarity information, which makes the noise rate lower. We also prove that not only the similarity labels but also the similarity noise transition matrix is robust to noise. Experiments are conducted on benchmark datasets, demonstrating the effectiveness of our method. In future work, investigating different types of noise for diverse real-life scenarios might prove important.

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

APPENDICES

## A  PROOF OF THEOREM 1

**Theorem 1.** *Assume that the dataset is balanced (each class has the same amount of samples, and $c$ classes in total), and the noise is class-dependent. Given a class noise transition matrix $T_c$, such that $T_{c,ij} = P(\bar{Y} = j | Y = i)$. The elements of the corresponding similarity noise transition matrix $T_s$ can be calculated as*

$$T_{s,00} = \frac{c^2 - c - \left( \sum_j (\sum_i T_{c,ij})^2 - ||T_c||_{\text{Fro}}^2 \right)}{c^2 - c}, \qquad T_{s,01} = \frac{\sum_j (\sum_i T_{c,ij})^2 - ||T_c||_{\text{Fro}}^2}{c^2 - c},$$

$$T_{s,10} = \frac{c - ||T_c||_{\text{Fro}}^2}{c}, \qquad\qquad\qquad T_{s,11} = \frac{||T_c||_{\text{Fro}}^2}{c}.$$

*Proof.* Assume each class has $n$ samples. $n^2 T_{c,ij} T_{c,i'j'}$ represents the number of sample-pairs generated by $(\bar{Y} = j | Y = i)$ and $(\bar{Y} = j' | Y = i')$. For the first element $T_{s,00}$, $n^2 \sum_{i \neq i'} T_{c,ij} T_{c,i'j'}$ is the number of sample-pairs with clean similarity labels $H = 0$, while $n^2 \sum_{i \neq i', j \neq j'} T_{c,ij} T_{c,i'j'}$ is the number of example-pairs with clean similarity labels $S = 0$ and noisy similarity labels $\bar{H} = 0$. Thus the ratio of these two terms is exact the $T_{s,00} = P(\bar{H} = 0 | H = 0)$. The remaining three elements can be represented in the same way. The primal representations are as follows,

$$T_{s,00} = \frac{\sum_{i \neq i', j \neq j'} T_{c,ij} T_{c,i'j'}}{\sum_{i \neq i'} T_{c,ij} T_{c,i'j'}}, \qquad T_{s,01} = \frac{\sum_{i \neq i', j = j'} T_{c,ij} T_{c,i'j'}}{\sum_{i \neq i'} T_{c,ij} T_{c,i'j'}},$$

$$T_{s,10} = \frac{\sum_{i = i', j \neq j'} T_{c,ij} T_{c,i'j'}}{\sum_{i = i'} T_{c,ij} T_{c,i'j'}}, \qquad T_{s,11} = \frac{\sum_{i = i', j = j'} T_{c,ij} T_{c,i'j'}}{\sum_{i = i'} T_{c,ij} T_{c,i'j'}}.$$

Further, note that

$$\sum_{i = i'} T_{c,i,j} T_{c,i',j'} = \sum_{i,j,j'} T_{c,i,j} T_{c,i,j'} = \sum_i (\sum_j T_{c,i,j})(\sum_{j'} T_{c,i,j'}) = c,$$

$$\sum_{i \neq i'} T_{c,i,j} T_{c,i',j'} = \sum_{i \neq i', j, j'} T_{c,i,j} T_{c,i',j'} = \sum_{i \neq i'} (\sum_j T_{c,i,j})(\sum_{j'} T_{c,i,j'}) = (c-1)c,$$

$$\sum_{i = i', j = j'} T_{c,ij} T_{c,i'j'} = ||T_c||_{\text{Fro}}^2,$$

$$\sum_{i \neq i', j = j'} T_{c,ij} T_{c,i'j'} = \sum_j (\sum_i T_{c,ij})^2 - ||T_c||_{\text{Fro}}^2.$$

Substituting above four equations to the primal representations, we have the Theorem 1 proved. □

## B  PROOF OF THEOREM 2

**Theorem 2.** *Assume that the dataset is balanced (each class has the same amount of samples), and the noise is class-dependent. When the number of classes $c \geq 8$, the noise rate for the noisy similarity labels is lower than that of the noisy class labels.*

*Proof.* Assume each class has $n$ samples. As we state in the proof of Theorem 1, the number of example-pairs with clean similarity labels $H = 0$ and noisy similarity labels $\bar{H} = 0$ is $n^2 \sum_{i \neq i', j \neq j'} T_{c,ij} T_{c,i'j'}$. We denote it by $N_{00}$. Similarly, we have,

$$N_{00} = n^2 \sum_{i \neq i', j \neq j'} T_{c,ij} T_{c,i'j'}, \qquad N_{01} = n^2 \sum_{i \neq i', j = j'} T_{c,ij} T_{c,i'j'},$$

$$N_{10} = n^2 \sum_{i = i', j \neq j'} T_{c,ij} T_{c,i'j'}, \qquad N_{11} = n^2 \sum_{i = i', j = j'} T_{c,ij} T_{c,i'j'}.$$

The noise rate is the ratio of the number of noisy examples to the number of total examples. Assume that the number of classes is $c$. We have

$$S_{noise} = \frac{N_{01} + N_{10}}{N_{00} + N_{01} + N_{10} + N_{11}} = \frac{N_{01} + N_{10}}{c^2 n^2},$$

$$C_{noise} = \frac{n \sum_{i \neq j} T_{c,ij}}{cn}.$$

Let $S_{noise}$ minus $C_{noise}$, we have

$$S_{noise} - C_{noise} = \frac{n^2 \sum_{i \neq i', j = j'} T_{c,ij} T_{c,i'j'} + n^2 \sum_{i = i', j \neq j'} T_{c,ij} T_{c,i'j'}}{c^2 n^2} - \frac{n \sum_{i \neq j} T_{c,ij}}{cn}$$

$$= \frac{\sum_{i \neq i', j = j'} T_{c,ij} T_{c,i'j'} + \sum_{i = i', j \neq j'} T_{c,ij} T_{c,i'j'} - c \sum_{i \neq j} T_{c,ij}}{c^2}.$$

Let $A = \sum_{i \neq i', j = j'} T_{c,ij} T_{c,i'j'} + \sum_{i = i', j \neq j'} T_{c,ij} T_{c,i'j'} - c \sum_{i \neq j} T_{c,ij}$, we have

$$A = \sum_{i \neq i', j = j'} T_{c,ij} T_{c,i'j'} + \sum_{i = i', j \neq j'} T_{c,ij} T_{c,i'j'} - c \sum_{i \neq j} T_{c,ij}$$

$$= \sum_{i \neq i', j = j'} T_{c,ij} T_{c,i'j'} + \sum_{i = i', j \neq j'} T_{c,ij} T_{c,i'j'} - c \left( \sum_{i,j} T_{c,ij} - \sum_{i = j} T_{c,ij} \right)$$

$$= \sum_{i \neq i', j = j'} T_{c,ij} T_{c,i'j'} + \sum_{i = i', j \neq j'} T_{c,ij} T_{c,i'j'} - c^2 + c \sum_{i = j} T_{c,ij}.$$

The second equation holds because the row sum of $T_c$ is 1.

For the first term $\sum_{i \neq i', j = j'} T_{c,ij} T_{c,i'j'}$, notice that:

$$\sum_{i \neq i', j = j'} T_{c,ij} T_{c,i'j'} = \sum_j \sum_i T_{c,ij} \left( \sum_{i' \neq i} T_{c,i'j} \right)$$

$$= \sum_j \sum_i T_{c,ij} \left( \sum_{i' \neq i} T_{c,i'j} + T_{c,ij} - T_{c,ij} \right)$$

$$= \sum_j \sum_i T_{c,ij} \left( \sum_{i'} T_{c,i'j} - T_{c,ij} \right)$$

$$= \sum_j \sum_i T_{c,ij} (S_j - T_{c,ij}) \qquad (S_j \text{ is the column sum of the } j - th \text{ column})$$

$$= \sum_j \sum_i T_{c,ij} S_j - T_{c,ij}^2$$

$$= \sum_j S_j \sum_i T_{c,ij} - \sum_j \sum_i T_{c,ij}^2$$

$$= \sum_j S_j^2 - \sum_j \sum_i T_{c,ij}^2. \qquad (6)$$

Due to the symmetry of $i$ and $j$, for the second term $\sum_{i = i', j \neq j'} T_{c,ij} T_{c,i'j'}$, we have

$$\sum_{i = i', j \neq j'} T_{c,ij} T_{c,i'j'} = \sum_j \sum_i T_{c,ij} (R_i - T_{c,ij}) \qquad (R_i \text{ is the row sum of the } i - th \text{ row, and } R_i = 1)$$

$$= \sum_j \sum_i T_{c,ij} - T_{c,ij}^2$$

$$= c - \sum_j \sum_i T_{c,ij}^2. \qquad (7)$$

Therefore, substituting Equation (6) and (7) into $A$, we have

$$A = \sum_j S_j^2 - \sum_j \sum_i T_{c,ij}^2 + c - \sum_j \sum_i T_{c,ij}^2 - c^2 + c \sum_{i = j} T_{c,ij}.$$

To prove $S_{noise} - C_{noise} \leq 0$ is equivalent to prove $A \leq 0$.

Let $M = c^2 - c$, $N = \sum_j S_j^2 - 2\sum_j \sum_i T_{ij}^2 + c\sum_{i=j} T_{ij}$ (we drop the subscript $c$ in $T_{c,ij}$), and $A = N - M$. Now we utilize the Adjustment method (Su & Xiong, 2015) to scale $N$. For every iteration, we denote the original $N$ by $N_o$, and the adjusted $N$ by $N_a$.

Since $c \geq 8$, there can not exist three columns with column sum bigger than $c/2 - 1$. Otherwise, the sum of the three columns will be bigger than $c$, which is impossible because the sum of the whole matrix is $c$.

Therefore, first, we assume that the $j, k - th$ columns have column sum bigger than $c/2 - 1$. Then, for the row $i$, we add the elements $l$, which are not in $j, k - th$ columns, to the diagonal element. We have

$$
\begin{aligned}
N_a - N_o &= (S_i + T_{il})^2 + (S_l + T_{il})^2 + cT_{il} - 2(T_{ii} + T_{il})^2 - S_i^2 - S_l^2 + 2(T_{ii}^2 + T_{il}^2) \\
&= T_{il}(2T_{il} + 2S_i - 2S_l + c - 4T_{ii}) \\
&\geq T_{il}(2T_{il} - 2S_l + c - 2T_{ii}) && (\because S_i \geq T_{ii}) \\
&> T_{il}(2T_{il} - c + 2 + c - 2T_{ii}) && (\because S_l < c/2 - 1) \\
&\geq 0. && (\because T_{ii} \leq 1)
\end{aligned}
$$

We do such adjustment to every rows, then $N_a$ is getting bigger and the adjusted matrix will only have values on diagonal elements and the $j, k - th$ columns. Since the diagonal elements are dominant in the row, $S_j + S_k < 2c/3 + 2/3$ (because for $i \neq j, k, T_{ij} + T_{ik} < 2/3$).

Assume that the column sum of $k - th$ column is no bigger than that of the $j - th$ column, and thus $S_k < c/3 + 1/3$. Then, for a row $i$, we add the $T_{ik}$ to $T_{ii}$. We have

$$
\begin{aligned}
N_a - N_o &= (S_i + T_{ik})^2 + (S_k + T_{ik})^2 + cT_{ik} - 2(T_{ii} + T_{ik})^2 - S_i^2 - S_k^2 + 2(T_{ii}^2 + T_{ik}^2) \\
&= T_{ik}(2T_{ik} + 2S_i - 2S_k + c - 4T_{ii}) \\
&\geq T_{ik}(2T_{ik} - 2S_k + c - 2T_{ii}) && (\because S_i \geq T_{ii}) \\
&> T_{ik}(2T_{ik} + c/3 - 2/3 - 2T_{ii}) && (\because S_k < c/3 + 1/3) \\
&\geq 0. && (\because c \geq 8, \text{ and } T_{ii} \leq 1)
\end{aligned}
$$

We do such adjustment to every rows, then $N_a$ is getting bigger and the adjusted matrix will only have values on diagonal elements and the $j - th$ column, which is called final matrix.

Note that if there is only one column with a column sum bigger than $c/2 - 1$, we can adjust the rest $c - 1$ columns as above and then obtain the final matrix as well. If there is no column with a column sum bigger than $c/2 - 1$, we can adjust all the elements as above and then obtain a *unit matrix*. For the unit matrix, $A = N - M < N_a - M = 0$, the Theorem 2 is proved.

Now we process the final matrix. For simplification, we assume $j = 0$ in the final matrix. We denote the $T_{ij}$ by $b_i$ and $T_{ii}$ by $a_i$, for $i = \{1, \ldots, c - 1\}$. We have

$$
\begin{aligned}
N_a &= \sum_i a_i^2 + (1 + \sum_i b_i)^2 + c(\sum_i a_i + 1) - 2(\sum_i a_i^2 + \sum_i b_i^2 + 1) \\
&= (1 + \sum_i b_i)^2 + c\sum_i a_i + c - \sum_i a_i^2 - 2\sum_i b_i^2 - 2 \\
&= 1 + (\sum_i b_i)^2 + 2\sum_i b_i + c\sum_i a_i + c - \sum_i a_i^2 - 2\sum_i b_i^2 - 2 \\
&= (\sum_i b_i)^2 + 2\sum_i b_i - 2\sum_i b_i^2 + c\sum_i a_i - \sum_i a_i^2 + c - 1 \\
&= (\sum_i b_i)^2 + 2\sum_i b_i - 2\sum_i b_i^2 + c\sum_i(1 - b_i) - \sum_i(1 - b_i)^2 + c - 1 \\
&= (\sum_i b_i)^2 + 2\sum_i b_i - 2\sum_i b_i^2 + c^2 - c - c\sum_i b_i - \sum_i(1 - 2b_i + b_i^2) + c - 1 \\
&= (\sum_i b_i)^2 + 4\sum_i b_i - 3\sum_i b_i^2 - c\sum_i b_i + c^2 - c.
\end{aligned}
$$

Now we prove $A = N - M \le N_a - M \le 0$. Note that

$$
\begin{aligned}
N_a - M &= (\sum_i b_i)^2 + 4\sum_i b_i - 3\sum_i b_i^2 - c\sum_i b_i \\
&= (\sum_i b_i)^2 + 3\sum_i b_i - 3\sum_i b_i^2 - (c-1)\sum_i b_i \\
&= (\sum_i b_i)^2 + 3\sum_i b_i - 3\sum_i b_i^2 - (\sum_i(1-b_i) + \sum_i b_i)\sum_i b_i \\
&= 3\sum_i b_i - 3\sum_i b_i^2 - \sum_i(1-b_i)\sum_i b_i \\
&= 3\sum_i b_i(1-b_i) - \sum_i(1-b_i)\sum_i b_i.
\end{aligned}
$$

According to the rearrangement inequality(Hardy et al., 1952), we have

$$
\sum_i(1-b_i)\sum_i b_i \ge (c-1)\sum_i b_i(1-b_i).
$$

Note that $c \ge 8$, thus $3\sum_i b_i(1-b_i) - \sum_i(1-b_i)\sum_i b_i \le 0$, and $A \le 0$. Therefore $S_{noise} - C_{noise} \le 0$, and the equation holds if and only if the noise rate is 0 or every instances have the same noisy class label (i.e., there is one column in the $T_c$, of which every elements are 1, and the rest elements of the $T_c$ are 0). Above two extreme situations are not considered in this paper. Namely, the noise rate of the noisy similarity labels is lower than that of the noisy class labels. Theorem 2 is proved.

$\square$

## C    PROOF OF THEOREM 3

**Theorem 3.** *Assume the parameter matrices $W_1, \ldots, W_d$ have Frobenius norm at most $M_1, \ldots, M_d$, and the activation functions are 1-Lipschitz, positive-homogeneous, and applied element-wise (such as the ReLU). Assume the transition matrix is given, and the instances $X$ are upper bounded by $B$, i.e., $\|X\| \le B$ for all $X$, and the loss function $\ell$ is upper bounded by $M$[3]. Then, for any $\delta > 0$, with probability at least $1 - \delta$,*

$$
R(\hat{f}) - R_n(\hat{f}) \le \frac{(T_{s,11} - T_{s,01})2BC(\sqrt{2d\log 2} + 1)\Pi_{i=1}^d M_i}{T_{s,11}\sqrt{n}} + M\sqrt{\frac{\log 1/\delta}{2n}}. \tag{8}
$$

*Proof.* We have defined

$$
R(f) = E_{(X_i, X_j, \bar{Y}_i, \bar{Y}_j, \bar{H}_{ij}, T_s) \sim \mathcal{D}_\rho}[\ell(f(X_i), f(X_j), T_s, \bar{H}_{ij})], \tag{9}
$$

and

$$
R_n(f) = \frac{1}{n^2}\sum_{i=1}^n\sum_{j=1}^n \ell(f(X_i), f(X_j), T_s, \bar{H}_{ij}), \tag{10}
$$

where $n$ is training sample size of the noisy data.

First, we bound the generalization error with Rademacher complexity (Bartlett & Mendelson, 2002).

**Theorem 4** (Bartlett & Mendelson (2002)). *Let the loss function be upper bounded by $M$. Then, for any $\delta > 0$, with the probability $1 - \delta$, we have*

$$
\sup_{f \in \mathcal{F}}|R(f) - R_n(f)| \le 2\mathfrak{R}_n(\ell \circ \mathcal{F}) + M\sqrt{\frac{\log 1/\delta}{2n}}, \tag{11}
$$

---

[3]The assumption holds because deep neural networks will always regulate the objective to be a finite value and thus the corresponding loss functions are of finite values.

where $\mathfrak{R}_n(\ell \circ \mathcal{F})$ is the Rademacher complexity defined by

$$\mathfrak{R}_n(\ell \circ \mathcal{F}) = E\left[\sup_{f \in \mathcal{F}} \frac{1}{n} \sum_{i=1}^{n} \sigma_i \ell(f(X_i), f(X_j), T_s, \bar{H}_{ij})\right], \tag{12}$$

and $\{\sigma_1, \cdots, \sigma_n\}$ are Rademacher variables uniformly distributed from $\{-1, 1\}$.

Before further upper bound the Rademacher complexity $\mathfrak{R}_n(\ell \circ \mathcal{F})$, we discuss the special loss function and its *Lipschitz continuity* w.r.t $h_k(X_i), k = \{1, \ldots, C\}$.

**Lemma 1.** *Given similarity transition matrix $T_s$, loss function $\ell(f(X_i), f(X_j), T_s, \bar{H}_{ij})$ is $\mu$-Lipschitz with respect to $h_k(X_i), k = \{1, \ldots, C\}$, and $\mu = (T_{s,11} - T_{s,01})/T_{s,11}$*

$$\left|\frac{\partial \ell(f(X_i), f(X_j), T_s, \bar{H}_{ij})}{\partial h_k(X_i)}\right| < \frac{T_{s,11} - T_{s,01}}{T_{s,11}}. \tag{13}$$

Detailed proof of Lemma 1 can be found in Section C.1.

Lemma 1 shows that the loss function is $\mu$-Lipschitz with respect to $h_k(X_i), k = \{1, \ldots, C\}$.

Based on Lemma 1, we can further upper bound the Rademacher complexity $\mathfrak{R}_n(\ell \circ \mathcal{F})$ by the following lemma.

**Lemma 2.** *Given similarity transition matrix $T_s$ and assume that loss function $\ell(f(X_i), f(X_j), T_s, \bar{H}_{ij})$ is $\mu$-Lipschitz with respect to $h_k(X_i), k = \{1, \ldots, C\}$, we have*

$$\mathfrak{R}_n(\ell \circ \mathcal{F}) = E\left[\sup_{f \in \mathcal{F}} \frac{1}{n} \sum_{i=1}^{n} \sigma_i \ell(f(X_i), f(X_j), T_s, \bar{H}_{ij})\right]$$

$$\leq \mu C E\left[\sup_{h \in H} \frac{1}{n} \sum_{i=1}^{n} \sigma_i h(X_i)\right], \tag{14}$$

*where $H$ is the function class induced by the deep neural network.*

Detailed proof of Lemma 2 can be found in Section C.2.

The right-hand side of the above inequality, indicating the hypothesis complexity of deep neural networks and bounding the Rademacher complexity, can be bounded by the following theorem.

**Theorem 5.** *(Golowich et al., 2018) Assume the Frobenius norm of the weight matrices $W_1, \ldots, W_d$ are at most $M_1, \ldots, M_d$. Let the activation functions be 1-Lipschitz, positive-homogeneous, and applied element-wise (such as the ReLU). Let $X$ is upper bounded by $B$, i.e., for any $X, \|X\| \leq B$. Then,*

$$E\left[\sup_{h \in H} \frac{1}{n} \sum_{i=1}^{n} \sigma_i h(X_i)\right] \leq \frac{B(\sqrt{2d\log 2} + 1)\Pi_{i=1}^{d} M_i}{\sqrt{n}}. \tag{15}$$

Combining Lemma 1,2, and Theorem 4, 5, Theorem 3 is proved. □

## C.1 PROOF OF LEMMA 1

Recall that

$$\begin{aligned}
\ell(f(X_i), f(X_j), T_s, \bar{H}_{ij} = 1) &= -\log(\hat{\bar{S}}_{ij}) \\
&= -\log(\hat{S}_{ij} \times T_{s,11} + (1 - \hat{S}_{ij}) \times T_{s,01}) \\
&= -\log(f(X_i)^{\top} f(X_j) \times T_{s,11} + (1 - f(X_i)^{\top} f(X_j)) \times T_{s,01}),
\end{aligned} \tag{16}$$

where

$$f(X_i) = [f_1(X_i), \ldots, f_c(X_i)]^\top$$
$$= \left[ \left( \frac{\exp(h_1(X))}{\sum_{k=1}^c \exp(h_k(X))} \right), \ldots, \left( \frac{\exp(h_c(X))}{\sum_{k=1}^c \exp(h_k(X))} \right) \right]^\top. \qquad (17)$$

Take the derivative of $\ell(f(X_i), f(X_j), T_s, \bar{H}_{ij} = 1)$ w.r.t. $h_k(X_i)$, we have

$$\frac{\partial \ell(f(X_i), f(X_j), T_s, \bar{H}_{ij} = 1)}{\partial h_k(X_i)} = \frac{\partial \ell(f(X_i), f(X_j), T_s, \bar{H}_{ij} = 1)}{\partial \hat{\bar{S}}_{ij}} \Big[ \frac{\partial f(X_i)}{\partial h_k(X_i)} \Big]^\top \frac{\partial \hat{\bar{S}}_{ij}}{\partial f(X_i)}, \quad (18)$$

where

$$\frac{\partial \ell(f(X_i), f(X_j), T_s, \bar{H}_{ij} = 1)}{\partial \hat{\bar{S}}_{ij}} = -\frac{1}{f(X_i)^\top f(X_j) \times T_{s,11} + (1 - f(X_i)^\top f(X_j)) \times T_{s,01}},$$

$$\frac{\partial \hat{\bar{S}}_{ij}}{\partial f(X_i)} = f(X_j) \times T_{s,11} - f(X_j) \times T_{s,01},$$

$$\frac{\partial f(X_i)}{\partial h_k(X_i)} = f'(X_i) = [f_1'(X_i), \ldots, f_c'(X_i)]^\top.$$

Note that the derivative of the softmax function has some properties, i.e., if $m \neq k$, $f_m'(X_i) = -f_m(X_i)f_k(X_i)$ and if $m = k$, $f_k'(X_i) = (1 - f_k(X_i))f_k(X_i)$.

We denote by $Vector_m$ the $m - th$ element in $Vector$ for those complex vectors. Because $0 < f_m(X_i) < 1, \forall m \in \{1, \ldots, c\}$, we have

$$f_m'(X_i) \leq |f_m'(X_i)| < f_m(X_i), \qquad\qquad \forall m \in \{1, \ldots, c\}; \qquad (19)$$
$$f'(X_i)^\top f(X_j) < f(X_i)^\top f(X_j). \qquad\qquad\qquad (20)$$

Therefore,

$$\left| \frac{\partial \ell(f(X_i), f(X_j), T_s, \bar{H}_{ij} = 1)}{\partial h_k(X_i)} \right| = \left| \frac{\partial \ell(f(X_i), f(X_j), T_s, \bar{H}_{ij} = 1)}{\partial \hat{\bar{S}}_{ij}} \Big[ \frac{\partial f(X_i)}{\partial h_k(X_i)} \Big]^\top \frac{\partial \hat{\bar{S}}_{ij}}{\partial f(X_i)} \right|$$

$$= \left| \frac{f'(X_i)^\top f(X_j) \times T_{s,11} - f'(X_i)^\top f(X_j) \times T_{s,01}}{f(X_i)^\top f(X_j) \times T_{s,11} + (1 - f(X_i)^\top f(X_j)) \times T_{s,01}} \right|$$

$$< \left| \frac{f(X_i)^\top f(X_j) \times T_{s,11} - f(X_i)^\top f(X_j) \times T_{s,01}}{f(X_i)^\top f(X_j) \times T_{s,11} + (1 - f(X_i)^\top f(X_j)) \times T_{s,01}} \right|$$

$$< \left| \frac{T_{s,11} - T_{s,01}}{T_{s,11}} \right|$$

$$= \frac{T_{s,11} - T_{s,01}}{T_{s,11}}. \qquad (21)$$

The second inequality holds because of $T_{s,11} > T_{s,01}$ (Detailed proof can be found in Section C.1.1) and Equation (20). The third inequality holds because of $f(X_i)^\top f(X_j) < 1$.

Similarly, we can prove

$$\left| \frac{\partial \ell(f(X_i), f(X_j), T_s, \bar{H}_{ij} = 0)}{\partial h_k(X_i)} \right| < \frac{T_{s,11} - T_{s,01}}{T_{s,11}}. \qquad (22)$$

Combining Equation (21) and Equation (22), we obtain

$$\left| \frac{\partial \ell(f(X_i), f(X_j), T_s, \bar{H}_{ij})}{\partial h_k(X_i)} \right| < \frac{T_{s,11} - T_{s,01}}{T_{s,11}}. \qquad (23)$$

### C.1.1 PROOF OF $T_{s,11} > T_{s,01}$

As we mentioned in Section B, we have,

$$N_{00} = n^2 \sum_{i \neq i', j \neq j'} T_{c,ij} T_{c,i'j'}, \qquad N_{01} = n^2 \sum_{i \neq i', j = j'} T_{c,ij} T_{c,i'j'},$$

$$N_{10} = n^2 \sum_{i = i', j \neq j'} T_{c,ij} T_{c,i'j'}, \qquad N_{11} = n^2 \sum_{i = i', j = j'} T_{c,ij} T_{c,i'j'},$$

$$T_{s,01} = \frac{N_{01}}{N_{00} + N_{01}}, \qquad T_{s,11} = \frac{N_{11}}{N_{10} + N_{11}},$$

$$T_{s,11} - T_{s,01} = \frac{N_{11} N_{00} + N_{11} N_{01} - N_{01} N_{10} - N_{01} N_{11}}{(N_{00} + N_{01})(N_{10} + N_{11})}.$$

Let us review the definition of similarity labels: if two instances belong to the same class, they will have similarity label $S = 1$, otherwise $S = 0$. That is to say, for a $k$-class dataset, only $\frac{1}{k}$ of similarity data has similarity labels $S = 1$, and the rest $1 - \frac{1}{k}$ has similarity labels $S = 0$. We denote the number of data with similarity labels $S = 1$ by $N_1$, otherwise $N_0$. Therefore, for the balanced dataset with $n$ samples of each class, $N_1 = cn^2$, and $N_0 = c(c-1)n^2$. Let $A = T_{s,11} - T_{s,01}$, we have

$$
\begin{aligned}
A &= N_{11} N_{00} - N_{01} N_{10} \\
&= N_{11} N_{00} - (N_0 - N_{00})(N_1 - N_{11}) \\
&= N_{11} N_{00} - N_0 N_1 - N_{11} N_{00} + N_{11} N_0 + N_1 N_{00} \\
&= N_{11} N_0 - N_{01} N_1 \\
&= c(c-1)n^2 N_{11} - cn^2 N_{01} \\
&> 0.
\end{aligned}
$$

The last equation holds because of $(c-1)N_{11} - N_{01} > 0$ according to the rearrangement inequality (Hardy et al., 1952).

### C.2 PROOF OF LEMMA 2

$$
\begin{aligned}
&E\left[\sup_{f \in \mathcal{F}} \frac{1}{n} \sum_{i=1}^{n} \sigma_i \ell(f(X_i), f(X_j), T_s, \bar{H}_{ij})\right] \\
&= E\left[\sup_{g} \frac{1}{n} \sum_{i=1}^{n} \sigma_i \ell(f(X_i), f(X_j), T_s, \bar{H}_{ij})\right] \\
&= E\left[\sup_{\operatorname{argmax}\{h_1, \ldots, h_C\}} \frac{1}{n} \sum_{i=1}^{n} \sigma_i \ell(f(X_i), f(X_j), T_s, \bar{H}_{ij})\right] \\
&= E\left[\sup_{\max\{h_1, \ldots, h_C\}} \frac{1}{n} \sum_{i=1}^{n} \sigma_i \ell(f(X_i), f(X_j), T_s, \bar{H}_{ij})\right] \\
&\leq E\left[\sum_{k=1}^{C} \sup_{h_k \in H} \frac{1}{n} \sum_{i=1}^{n} \sigma_i \ell(f(X_i), f(X_j), T_s, \bar{H}_{ij})\right] \\
&= \sum_{k=1}^{C} E\left[\sup_{h_k \in H} \frac{1}{n} \sum_{i=1}^{n} \sigma_i \ell(f(X_i), f(X_j), T_s, \bar{H}_{ij})\right] \\
&\leq \mu C E\left[\sup_{h_k \in H} \frac{1}{n} \sum_{i=1}^{n} \sigma_i h_k(X_i)\right] \\
&= \mu C E\left[\sup_{h \in H} \frac{1}{n} \sum_{i=1}^{n} \sigma_i h(X_i)\right],
\end{aligned}
$$

where the first three equations hold because given $T_s$, $f$ and $\max\{h_1, \ldots, h_C\}$ give the same constraint on $h_j(X_i), j = \{1, \ldots, C\}$; the sixth inequality holds because of the Talagrand Contraction Lemma (Ledoux & Talagrand, 2013).

## D    DEFINITION OF NOISE SETTINGS

Symmetric noise setting is defined as follows, where $C$ is the number of classes.

$$\text{Sym-}\rho: \quad T = \begin{bmatrix} 1-\rho & \frac{\rho}{C-1} & \cdots & \frac{\rho}{C-1} & \frac{\rho}{C-1} \\ \frac{\rho}{C-1} & 1-\rho & \frac{\rho}{C-1} & \cdots & \frac{\rho}{C-1} \\ \vdots & & \ddots & & \vdots \\ \frac{\rho}{C-1} & \cdots & \frac{\rho}{C-1} & 1-\rho & \frac{\rho}{C-1} \\ \frac{\rho}{C-1} & \frac{\rho}{C-1} & \cdots & \frac{\rho}{C-1} & 1-\rho \end{bmatrix}. \tag{24}$$

The 0.3 asymmetric noise setting is set as follow,

```
def asym_transition_matrix_generate(noise_rate=0.3, num_classes=10,
seed=1):
np.random.seed(seed)
t = np.random.rand(num_classes, num_classes)
i = np.eye(num_classes)
t = t + 1.2 * num_classes * i
for a in range(num_classes):
t[a] = t[a] / t[a].sum()

return t
```

Listing 1: Asymmetric noise (transition matrix) generation.

## E    EXPERIMENTS

### E.1    GIVEN GROUND-TRUTH CLASS $T_c$

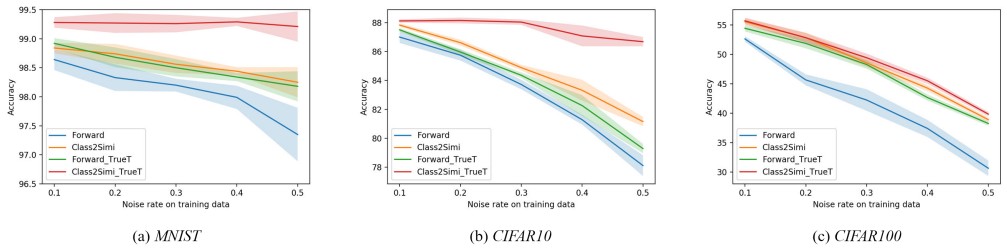

(a) *MNIST*                    (b) *CIFAR10*                    (c) *CIFAR100*

Figure 5: Means and Standard Deviations (Percentage) of Classification Accuracy over 5 trials on *MNIST*, *CIFAR10* and *CIFAR100* with symmetric noise. Forward and Class2Simi are trained with estimated T. Forward_TrueT is trained with ground-truth class $T_c$ and Class2Simi_TrueT is trained with similarity $T_s$ calculated from ground-truth class $T_c$.

From Figure 5, overall, we can see that Class2Simi (Class2Simi_TrueT) achieves the best performance whenever class $T_c$ is given or estimated. In most cases, Class2Simi with estimated $T_c$ even outperforms baselines with the ground-truth class noise transition matrix, due to lower noise rate and the similarity transition matrix being robust to noise. Specifically, On *MNIST*, as the noise rate increases from Sym-0.1 to Sym-0.5, Class2Simi_TrueT maintains remarkable accuracy above 99.20% while the accuracy of Class2Simi and Forward_TrueT decrease steadily. However, there is a significant decrease in the accuracy of Forward. On *CIFAR10*, the patterns of varying tendencies of four curves are similar to that of *MNIST* except that the decreases are more dramatic and even Class2Simi_TrueT drops slightly at Sym-0.5. On *CIFAR100*, there is an obvious decrease in the accuracy of all methods and our method achieves the best results across all noise rate, i.e., at Sym-0.5, Class2Simi gives an accuracy uplift of about 8.0% compared with Forward.

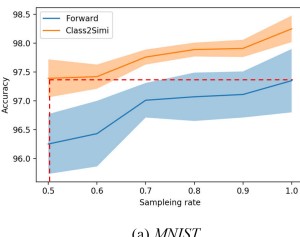 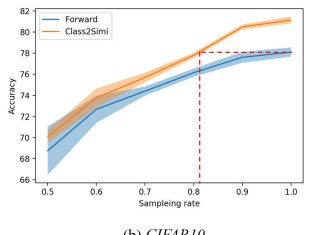 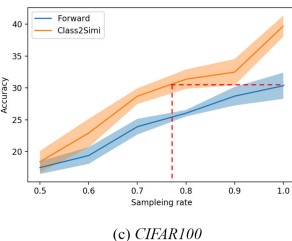

(a) *MNIST*              (b) *CIFAR10*              (c) *CIFAR100*

Figure 6: Means and Standard Deviations (Percentage) of Classification Accuracy over 5 trials on *MNIST*, *CIFAR10* and *CIFAR100* trained with different sampling rate of training data. The noise rate on training data is set to Sym-0.5.

## E.2 TRAINING WITH DIFFERENT SAMPLING RATE ON TRAINING DATA

In Figure 6, we show that the Class2Simi performs well even with fewer data. The noise in training data is set to Sym-0.5. We randomly sample from original training data with a sampling rate from 0.5 to 1.0 and train the model on the sampled data. Test datasets remain the same. At each sampling rate, Class2Simi performs better than the baseline. With only 50%, 80% and 80% data on *MNIST*, *CIFAR10* and *CIFAR100*, our method can achieve the same accuracy as Forward.

## E.3 RESULTS ON ASYMMETRIC NOISE SETTING

Table 5: Means and Standard Deviations (Percentage) of Classification Accuracy over 5 trials on *MNIST*, *CIFAR10* and *CIFAR100* with asymmetric noise of which the noise rate is about 0.3.

| 0.3 Asymmetric Noise | *MNIST* | *CIFAR10* | *CIFAR100* |
|---|---|---|---|
| Co-teaching | 97.99±0.16 | 83.08±0.22 | 47.07±0.84 |
| APL | 98.69±0.14 | 80.99±0.79 | 28.28±1.69 |
| S2E | 97.98±0.06 | 57.02±1.46 | 43.67±1.48 |
| Forward | 98.30±0.33 | 84.03±0.47 | 42.77±1.52 |
| Forward & Class2Simi | **98.44±0.14** | **85.05±0.72** | **49.96±0.88** |
| Reweight | 97.70±0.12 | 84.10±0.32 | 43.07±1.01 |
| Reweight & Class2Simi | **97.76±0.38** | **84.63±0.19** | **48.47±0.85** |
| Revision | 98.21±0.11 | 84.85±0.35 | 43.14±0.98 |
| Revision & Class2Simi | **98.24±0.15** | **85.11±0.27** | **48.69±0.83** |

In Table 5, we demonstrate the effectiveness of our method under the asymmetric noise setting.

## F LEARNABILTY CONNECTION BETWEEN PAIRWISE CLASSIFICATION AND POINTWISE CLASSIFICATION

### F.1 *Pointwise* IMPLIES *pairwise*

For an invertible $T_c$, denote by $\mathbf{v}_j$ the $j$-th column of $T_c$ and $\mathbf{1}$ the all-one vector. Then,

$$\sum_j (\sum_i T_{c,ij})^2 = \sum_j \langle \mathbf{v}_j, \mathbf{1} \rangle^2 \leq \sum_j ||\mathbf{v}_j||^2 ||\mathbf{1}||^2 = c ||T_c||_{\mathrm{Fro}}^2,$$

where we use the Cauchy–Schwarz inequality (Steele, 2004) in the second step. Further, we have

$$
\begin{aligned}
T_{s,11} + T_{s,00} &= \frac{||T_c||^2_{\text{Fro}}}{c} + \frac{c^2 - c - \left(\sum_j (\sum_i T_{c,ij})^2 - ||T_c||^2_{\text{Fro}}\right)}{c^2 - c} \\
&= \frac{(c-1)||T_c||^2_{\text{Fro}} + c^2 - c - \left(\sum_j (\sum_i T_{c,ij})^2 - ||T_c||^2_{\text{Fro}}\right)}{c^2 - c} \\
&= \frac{(c-1)||T_c||^2_{\text{Fro}} + c^2 - c - \left(\sum_j \langle \mathbf{v}_j, \mathbf{1}\rangle^2 - ||T_c||^2_{\text{Fro}}\right)}{c^2 - c} \\
&\geq \frac{(c-1)||T_c||^2_{\text{Fro}} + c^2 - c - \left(c||T_c||^2_{\text{Fro}} - ||T_c||^2_{\text{Fro}}\right)}{c^2 - c} \\
&= 1.
\end{aligned}
$$

Thus the learnability of the pointwise classification implies the learnability of the reduced pairwise classification.

### F.2  *Pairwise* DOES NOT IMPLY *pointwise*

Assuming the number of classes is 10 and the dataset is balanced, a singular $T_c$ of the pair noise pattern and the transformed $T_s$ are shown as follows,

$$
T_c = \begin{bmatrix}
0.5 & 0.5 & \dots & 0 & 0 \\
0 & 0.5 & 0.5 & \dots & 0 \\
\vdots & & \ddots & & \vdots \\
0 & \dots & 0 & 0.5 & 0.5 \\
0.5 & 0 & \dots & 0 & 0.5
\end{bmatrix}, \qquad
T_s = \begin{bmatrix}
0.95 & 0.05 \\
0.50 & 0.50
\end{bmatrix}.
$$

In this case, the reduced pairwise classification is learnable while the original pointwise classification is not learnable.

Thus the learnability of the reduced pairwise classification does not imply the learnability of the pointwise classification.

