# OpenReview forum: "Class2Simi: A New Perspective on Learning with Label Noise"
_ICLR.cc/2021/Conference — Reject_

### Official Review · AnonReviewer3 · 2020-10-28
**This work presents a new technique to address noise in labels by converting class labels to similary labels which has been shown to be very effective in noisy environment.**

**Rating:** 5
**Confidence:** 4

**Review:**

In this work the authors propose a method to learn from noisy labels. The propose method converts the noisy labels to similarity labels which are more robust to noise and helps in reducing the noise ratio in the training data. The proposed method has been evaluated on multiple datasets with consistent improvement across all noise ratios.

Pros:

The proposed idea to convert the class labels to similarity labels is very interesting and intuitive. It has some good properties which leads to a better performance.

The authors provide a theoretical analysis of the proposed method to estimate similarity noise transition matrix which makes it more grounded.

The authors have provided sufficient experimental results on multiple datasets to demonstrate effectiveness of the proposed learning technique.

Cons:

For asymmetric noise, the results are shown only for a noise rate of 0.3. How will the method perform when the noise rate is lower or higher in case of asymmetric noise. Also, how does the proposed method compares with existing approaches for asymmetric noisy learning? The asymmetric noisy learning in Li et. al. 2019 (and many others) can achieve around 93% accuracy on CIFAR-10 which is 10% higher then the proposed method.

The presented ablation study is not very meaningful. The authors have shown that on a clean dataset, the proposed method does not have any effect, sometimes the score improves, sometimes it goes down. Ideally the performance should not have been gone down, how will the authors explain the results on news20?

-- post rebuttal --

After carefully reading the authors response and the discussions with other reviewers, I am updating my ratings. The authors acknowledged that it is not a theory paper and therefore, the biggest concern is empirical evaluation. The shown performance is not comparable with existing high performing methods and therefore it is hard to judge without any direct comparison. The authors stated that their method can be applied on top of any existing method, which was not shown in this submission and therefore it will not be meaningful to judge just based on this statement. I believe showing this empirically will definitely strengthen this submission.

---

> ### Author Response · Authors · 2020-11-18
> **Thanks for the valuable feedback.  We will carefully address the concerns one by one.**
>
> **Q1:** How will the method perform when the noise rate is lower or higher in case of asymmetric noise.
>
> **A1:** We conducted experiments on CIFAR10 and CIFAR100 with asymmetric noise rates at 0.1 and 0.5. Means and standard deviations (percentage) of classification accuracy over 5 trials are shown in the below table. We can see that when the noise rate is lower or higher in the case of asymmetric noise, our method is still effective.
>
> | CIFAR10 with asymmetric noise | 0.1                | 0.3                | 0.5                |
> | ----------------------------- | ------------------ | ------------------ | ------------------ |
> | Forward                       | 86.68$\pm$0.07     | 84.65$\pm$0.39     | 79.04$\pm$0.90     |
> | Forward & Class2Simi          | **86.71$\pm$0.15** | **85.32$\pm$0.16** | **80.85$\pm$0.30** |
>
> | CIFAR100 with asymmetric noise | 0.1                | 0.3                | 0.5                |
> | ------------------------------ | ------------------ | ------------------ | ------------------ |
> | Forward                        | 49.90$\pm$0.56     | 42.77$\pm$1.52     | 31.34$\pm$0.91     |
> | Forward & Class2Simi           | **53.07$\pm$0.57** | **49.96$\pm$0.88** | **33.73$\pm$1.28** |
>
>
> As for the SOTA paper DivideMix, it is a very powerful algorithm against label noise, yet it is a little complex. It introduces a **two-component Gaussian Mixture Model (GMM)** to distinguish clean and noisy samples. It exploits **MixMatch** for SSL. It uses **MixUp** augmentation. **A regularization term** is used to prevent assigning all samples to a single class. There is also a **special penalty for asymmetric noise**.
>
> As for our method, we simply add a similarity transition matrix before the modified c2s loss function. In the experiments, for all noise settings, we use the same hyper-parameters without careful design. On the clean CIFAR10 dataset, the model (including the network, optimizer, and hyper-parameters) we used can only achieve an accuracy of 89.18 as shown in the ablation study. Based on this, the effectiveness of our method against label noise is significant.
>
> **Q2:** The presented ablation study is not very meaningful.
>
> **A2:** Thank you! Do you have any suggestions on the ablation study? We are happy to do more research to make the ablation study more sufficient and convincing.
>
> The ablation study we have done is to investigate how the similarity loss function influences the classification accuracy.  Specifically, the three T-based methods used the cross-entropy loss function while class2simi used the similarity loss function. We believe that it is necessary to do such an ablation study.
>
> On clean News20, the CE loss function performs better than the similarity loss function, while on noisy News20, the results reverse, which is much powerful to support the effectiveness of our method.

---

> > ### Comment · AnonReviewer3 · 2020-11-24
> > **Additional questions on performance**
> >
> > I would like to thank the authors for their effort in answering the questions.
> >
> > The authors have indicated that this is not a theory paper, which puts more emphasis on the empirical performance. Regarding the performance comparison with existing methods, such as DivideMix, it is not sufficient to merely state that this method is more complex. The performance gap is not a small margin, it is a very significant difference. It seems, the proposed approach is not general enough to be applied on top of DivideMix or any other noise handling approach. It would help if the authors can comment on this and explain how the proposed approach can be useful in addressing noisy learning from practical point of view.

---

> > > ### Author Response · Authors · 2020-11-24
> > > **Class2Simi is a meta method for handling label noise**
> > >
> > > In fact, it is possible to apply our method on top of DivideMix.
> > >
> > > Let us first analyze what was done in DivideMix and/or similar methods. This family has two components: a **sample selector** to divide the set of noisy data into a set of intact data and a set of mislabeled data, where the latter will be regarded as a set of unlabeled data; a **semi-supervised learner** (either graph-based or perturbation-based) to recover the true labels of those unlabeled data (either model-based or model-free). Combining the two components, we can obtain a **great label corrector**. A different combination of the sample selector and semi-supervised learner (from DivideMix) can result in a different yet still great label corrector.
> > >
> > > If the sample selector is perfect, the precision is 1 (all selected data are correctly labeled) and the recall is 1 (all correctly labeled data are selected). If the semi-supervised learner is perfect, all unlabeled data can be correctly labeled. Whenever **one of them is imperfect**, the processed data remain some amount of label noise (though significantly reduced), and there is some room to apply Class2Simi to further reduce the label noise (the noise reduction and learnability are theoretically guaranteed).
> > >
> > > As we replied to AC, given any **Base** handling label noise, **Class2Simi is applicable on top of Base**. In one way, we can replace Forward/Reweight/Revision with Base after transforming noisy class labels to noisy pairwise labels. In the other way, if Base is sample selection or label correction, we can first apply Base to reduce noise once and then transform noise-reduced noisy class labels to noisy pairwise labels to reduce noise once more. No matter how small the amount of label noise remains after being processed by Base, the amount of label noise should be smaller after being transformed into pairwise data.
> > >
> > > In this sense, Class2Simi is a **meta** method for handling label noise.
> > >
> > > Thanks!

---

> > > ### Author Response · Authors · 2020-11-24
> > > **We are still running more experiments**
> > >
> > > DivideMix is definitely **a genius idea** for introducing semi-supervised learning into label-noise learning. By doing so, it incorporates many powerful augmentation techniques (including MixUp and Virtual Adversarial Training inside MixMatch). We would like to see the performance of Class2Simi if we **turn on** the standard and advanced augmentations. However, we don't have the computational resources to finish the ablation study within one/two weeks.
> > >
> > > BTW, we originally thought our work was more empirical than theoretical. After very long discussions about the **learnability, calibration, and bounds**, we realized that our work is more theoretical than empirical!

---

### Official Review · AnonReviewer4 · 2020-10-28
**A novel model with good experimental support but marginal technical novelty**

**Rating:** 6
**Confidence:** 4

**Review:**

This paper presented a working framework for learning a robust classifier with noisy labels. It proves that if the number of the classes is more than 8 then the noise rate in the similarity matrix is less than that in the noise rate in labels. Hence after learning classifiers from noisy labels, it updates the classifier using entropy loss between predicted similarity matrix and actual similarity matrix calculated from similarity matrix from data multiplied with noisy transition matrix derived using existing techniques(Xia et al., 2019; Patrini et al., 2017).

It also provides the generalization bound for the proposed techniques.
The paper has a strong experimental evaluation of the proposed method against recent models on robust learning with noisy labels.

My concern is the technical novelty of the proposed model, as learning from the noise transition matrix (Xia et al., 2019; Patrini et al., 2017) and learning from the similarity matrix (Hsu et al., 2019) both are well known. It will be better if the author discusses their contribution in detail.


It will be good if the authors also comment on the running time of the proposed method as it is learning the classifier first from the noisy labels. What happens if one learns ‘f’ in stage 2 with a random classifier?

In Tabel 1 for most of the cases, forward+C2S is outperforming others. Why? In Table 2, we have seen that model does not make a significant change in the news20 data set. Why so?

We have seen that difference in performance is higher with a higher noise rate in the labels. The authors have stated in Theorem 1 that the noise rate for the noisy similarity labels is lower than that of the noisy class labels. But they did not provide any quantitative analysis of that. It will be good to see what is the reduction in the rate with respect to the number of classes and also the noise rate in labels.

I tend to accept this paper with clarification asked. Though the paper has combined existing ideas to learn a robust classifier, the proposed classifier is outperforming when the noise rate is higher. Hence it can be useful for the ML communities. Along with this, the paper has also given a generalization bound.


---------------------
I thank the author for clarification. I would like to be with my score.

---

> ### Author Response · Authors · 2020-11-16
> **Thanks for the valuable feedback.  We will carefully address the concerns one by one.**
>
> **Q1:** My concern is the technical novelty of the proposed model, as learning from the noise transition matrix (Xia et al., 2019; Patrini et al., 2017) and learning from the similarity matrix (Hsu et al., 2019) both are well known. It will be better if the author discusses their contribution in detail.
>
> **A1:**  First, we propose a new perspective on learning with label noise, which transforms class labels into similarity labels. Such transformation not only reduces the noise level but also introduces the transition matrix more robust against estimation errors. The new perspective is novel and effective.
>
> Theoretically, we proved the noise rate is reduced and we gave a generalization error bound.
>
> Technically, we provide a way to estimate the **similarity noise transition matrix** by theoretically establishing its relation to class noise transition matrix
>
> **Q2:** Running time. What happens if one learns ‘f’ in stage 2 with a random classifier?
>
> **A2:** Compared with learning from clean data, estimating noise transition matrix (noise rate) do increase the computational cost, as well as running time. However, this step is **necessary** in most label noise learning algorithms [1] [2]. In practice, it increases less than 15% of running time. For example, on CIFAR100, we train the model 200 epochs in stage 2 while we only train the model 20 epochs to estimate the noise rate.
>
> If we learns ‘f’ in stage 2 with a random classifier, as we stated in Section 3.3 implement, the classifier will lose the mapping between the output nodes and the semantic classes. This is a limitation of learning only from similarity labels, and we address this issue by  loading the model in stage 1.
>
> [1] Patrini, Giorgio, et al. "Making deep neural networks robust to label noise: A loss correction approach." *CVPR*. 2017.
>
> [2] Xia, Xiaobo, et al. "Are Anchor Points Really Indispensable in Label-Noise Learning?." *NIPS*. 2019.
>
>
> **Q3:** In Tabel 1 for most of the cases, forward+C2S is outperforming others. Why? In Table 2, we have seen that model does not make a significant change in the news20 data set. Why so?
>
> **A3:** The comparison should be internal, like forward vs. forward+c2s, and reweight vs. reweight+c2s. We can see that the improvements are similar among three T-based methods. As forward outperforms others,  forward+c2s should also outperforms others.
>
> MNIST, CIFAR10/100 are image datasets containing adequate instances. However, News20 is a text dataset containing only 20,000 instances of 20 classes. Although we use GloVe to extract features from raw text data, the representations are not good as those extracted by CNN from image data. Thus the model is hard to fit the noisy data, which weakens the effect of noisy to the classifier to some extent. From Table1, 2 and Table 4, which shows the results on clean data, we can see the gap between the accuracy on clean data and the accuracy on noisy is the smallest for News20.
>
>
>
> **Q4:**  It will be good to see what is the reduction in the rate with respect to the number of classes and also the noise rate in labels.
>
> **A4:** Thanks for raising this insightful point. Up to now, given a noisy dataset, the exact value of reduction can be calculated since all the related information is determined. The mathematical formula of the reduction with respect  to the number of classes and also the noise rate is **contained** in the proof of Theorem 2 (in the bottom of Page 12), which is very complex. Without additional assumptions, it is hard to simplify the formula because the formula is associated with every element in the transition matrix. It is still interesting to see how the number of classes and also the noise rate contributes.

---

### Official Review · AnonReviewer1 · 2020-10-29
**The idea is novel. Deeper analysis is needed for experiment.**

**Rating:** 6
**Confidence:** 4

**Review:**

This paper proposes a new perspective on dealing with label noise, called Class2Simi, by transforming the training examples with noisy labels into pairs of examples with noisy similarity labels and then learning a deep model with the noisy similarity labels. Experimental results on real datasets show that Class2Simi achieves better classification accuracy than its baselines that directly deals with the noisy class labels.

The idea to deal with label noise by transforming noisy class labels into noisy similarity labels seems to be novel. The proposed Class2Simi provides a framework to improve different existing learning methods. Furthermore, the paper proves that the noise rate for the noisy similarity labels is lower than that of the noisy class labels. In addition, the paper is well written with good organization.

Although in most cases the proposed Class2Simi can improve the accuracy compared with baselines, the improvement is not significant in many cases like those on MNIST and CIFAR10. It is better to provide deep analysis about the principle of the proposed method and the experimental results, and give insight for readers about when the proposed method will achieve significant improvement and what is the underlying reason.

-------------------
After rebuttal:
I thank the authors for clarification. I would like to keep with my score.

---

> ### Author Response · Authors · 2020-11-16
> **Thanks for the valuable feedback.  We will carefully address the concerns one by one.**
>
> **Q1:**  It is better to provide deep analysis about the principle of the proposed method and the experimental results, and give insight for readers about when the proposed method will achieve significant improvement and what is the underlying reason.
>
> **A1:** We thank your valuable feedback. In the ablation study, we found the similarity loss function does not improve the classification accuracy. Thus the accuracy increase is benefited from the two main advantages of the Class2Simi transformation. The first one is that the noise rate gets lower, which has been theoretically proved. The second one is that the similarity noise transition matrix is more robust against estimation errors , which has been empirically verified, as shown in Figure 4. We think the improvement varying on different datasets is due to different data structures, such as the number of class, the inherent similarity between different classes.

---

### Official Review · AnonReviewer2 · 2020-11-03
**Original and interesting idea but several major concerns regarding clarity, theory, and experiments.**

**Rating:** 3
**Confidence:** 4

**Review:**

**Summary**
The paper addresses the problem of learning with noisy labels by transforming the original category classification task into a semantic-similarity prediction task. The new task takes pairs of samples as input and predicts if the two samples are coming from the same category or not. It is theoretically shown that the similarity pairs have lower noise rates compared to the original categorization. It is then argued that this lower noise rate makes the learning more robust in the presence of class label noise.
In practice, the idea is applied on top of a standard categorization network (equipped with other approaches to handle class label noise). Specifically, for each pair, a similarity score $\in[0,1]$ is obtained from the category outputs of the two samples as the dot product of their softmax distributions. Then, a similarity transition matrix (that is pretrained) is applied to the prediction. Finally, a two-class cross-entropy loss is optimized.
The similarity transition matrix is obtained by using prior techniques for finding a $C\times C$ class transition matrix (with $C$ being the number of classes) and then analytically turning it into a $2\times2$ similarity transition matrix.

**Quality**
The writing is of a noticeably-low quality and seems to have been overly-rushed. The experiments are done on several datasets of different modalities. The theories are partially informative but not entirely and/or directly relevant.

**Clarity**
The method section is not clearly written. Section 3.1 and 3.2 were quite hard to follow due to some notational inconsistencies, lacking definitions, and not being self-contained (relies on the knowledge of Hsu et al. 2018). The relevance and implications of the theories are not properly discussed.

**Originality**
The idea of turning category classification to similarity prediction in order to make the learning robust to class label noise seems quite original to the reviewer’s knowledge.

**Significance**
The results show improvements on various benchmarks as well as various underlying category classification methods for learning under noisy labels. The improvements are not always substantial but they seem to be statistically significant and consistently present for large synthetic symmetric noise levels of CIFAR100. The idea being general and the results somewhat significant indicate a potential significance of the method.

**Major technical comments**

*Theory*

1. when learning with similarity labels, is it important to consider the total noise levels among similar and dissimilar pairs or also the worst case of noise level in similar pairs and dissimilar pairs separately? To make this more clear, imagine the extreme case that all of the given labels are noisy. Even for this case, when constructing the similarity labels, while all similar labels can be wrong, still the majority of dissimilar labels (and vast majority if the number of classes are high) will be correct. This renders the similarity noise rate to be (arbitrarily) low (depending on the number of classes and samples per classes). Does this low noise rate mean the task is learnable although there is literally zero information on true class labels? I doubt it. That would essentially mean we can take any set of images and apply random similar/dissimilar pairs to them and consider it a low-noise-rate dataset. Thus, formal analysis and/or informal theoretical discussions are required in this regard.
2. from algorithm 1 it seems the method requires two independent trainings, shouldn’t that at least double the computational complexity? Learning with pairs could potentially take longer to converge due to the quadratic increase in the number of input data. How is the claim in section 3.3 that “Class2Simi increases the computation cost *slightly*” supported?
3. regarding theorem 3, based on the first point above, I have a concern that the relevant risk to be studied here should be the original category classification risk as opposed to the similarity risk. The latter could be dominated by the dissimilar pairs and unless formally analyzed or at least directly discussed it’s hard to draw any conclusion on how the bound on the similarity prediction empirical risk translates to a bound on the original classification risk which is the objective of interest.

*Experiments*

1. Regarding the experiments on Clothing1M, while I understand the raised points regarding the dominant class confusion, I believe the method should still be compared on the original dataset along with the proposed Clothing1M*. In fact, this can be a weakness of the proposed method that should be studied further and more thoroughly with designated experiments. Such a noise is possible in the real-world applications due to semantic ambiguity or human error (as demonstrated in Clothing1M).
2. Is the same set of hyperparameters used for all the baselines as well as the variants of the proposed approach? How are the hyperparameters optimizations done? In particular, which variant of the method or baselines are the hyperparameters optimized on? In our experience, when it comes to noisy labels, it is quite common that different methods perform better with different sets of hyperparameters, so it’s important to optimize the parameters per method.
3. the improvements for asymmetric noise, and on MNIST and CIFAR seem marginal. A statistical paired significance test could be useful.

**Minor technical comments**

- the notation of the summands’ subindices $i,j,i’,j’$, in theorem 1 is a bit confusing. For instance, when $i=i’$ should only $i$ be used?
- the similarity label is denoted as $H_{ij}$ in section 3.1but as $S_{i,j}$ in figure 2 and section 3.2.
- what is the difference between samples denoted by $X_i$ in section 3.2 and $x_i$ in section 3.1?
- what does $\theta$ parametrize? The network or the similarity transition matrix? What is the difference between $f(X_i)$ and $f(X_i;\theta)$ in figure 2?
- the algorithm defines a function $g(.)$ which is not referred to in the methods section
- where are parameter matrices $W_1,...,W_d$ defined? What is d?
- how is the expected risk $R$ defined?
- what is $\hat{f}$
- what is $R_n$?

**Overall**
While the reviewer can guess the meaning of some of the notations based on the literature it renders the paper hardly readable and at times the discrepancies were unresolvable to the reviewer. Furthermore, there are concerns regarding the motivation, relevance of the theories, and the significance of the results. So, overall, while I believe the paper has original contributions of potentially high impact I strongly believe it needs a major revision before it’s presentable at a conference.

**Post Rebuttal**
I had concerns regarding the clarity, theory, and experiments. During the rebuttal phase, the authors actively discussed various points raised by the reviewers and the AC. Despite its length and breadth, I do not find them addressing the core of the raised concerns. That is except my 2nd point of the major theory concerns, regarding the time complexity and convergence time which is at least partially addressed.  The time complexity is addressed since the length of the first round of training for transition matrix is negligible compared to the main training. The convergence time would also be addressed if the number of epochs for the proposed method is the same as the baselines. The reviewer is not entirely sure that is the case though. Given the outstanding majority of the concerns my final feedback is as follows:

The paper provides an original idea for learning with label noise which is positive, to rate the demonstration of the relevance of the idea, I can either consider the paper from an empirical study lens or a theoretical one.

From the latter perspective, the concerns above effectively affect the whole theoretical arguments of the paper. The main claim of the paper, even in the latest revision, is based on the noise rate of similarity labels being lower than the noise rate of the corresponding class labels and that this is what can bring improvement in the final performance. This reads absolutely unfounded to the reviewer. As discussed, the similarity rate is mostly influenced by dissimilar labels (in the balanced c>2-classification scenario) and can be made arbitrarily low. Furthermore, the discussion still does not make a clear formal connection between the error bound on the noisy similarity learning and the noisy classification for the general case. Nevertheless, such a connection would require a major revision/addition to the paper that would need a proper round of review.

When it comes to the empirical view, the experiments are inconclusive and not thorough-enough for an empirical paper due to 1) the drastic change in the learning setup (which is implemented inhouse including the base transition-matrix methods) in tandem with the fact that hyperparameter optimization is not done per method (e.g., the hyperparameters are taken from the papers for baselines while they are optimized for the proposed method's training). This is especially important since when learning with noisy labels, the choice of hyperparameters are extremely influential in the final results. 2) the improvements are mostly marginal except for CIFAR100. 3) Results are not provided for the original Clothing1M dataset. The shortcoming that led to changing Clothing1M needs to be thoroughly studied for an empirical work since it directly affects the applicability of the paper.

On top of these, the final version of the pdf is still lacking on clarity several instances of which were listed in the original review.

Thus, considering all the points above leads me to confidently keep the original rating as "clear reject".

---

> ### Author Response · Authors · 2020-11-16
> **Thanks for the valuable feedback.  We will carefully address the concerns one by one.**
>
> **Q1:** To make this more clear, imagine the extreme case that all of the given labels are noisy...... Does this low noise rate mean the task is learnable although there is literally zero information on true class labels? I doubt it.
>
> **A1:**  The extreme case you took up is that all of the given labels are noisy. To make this more clear, if the "**noisy**" means that the all the labels of instances **may not** be the ground-truth labels, this case is common in label noise settings. Namely, all the training data is noisy, and there is no additional clean data to facilitate the training. This is the case we considered, and the task is learnable because there is literally **some** information on true class labels with the assumption that true class is dominant. In this case, the class2simi transformation can reduce the noise rate as we proved, and the experimental results showed the effectiveness.
>
> If the "**noisy**" means that the all the labels of instances **must not** be the ground-truth labels, i.e., all the labels are incorrect. In this case, the noise rate is 100%. The transformation definitely can reduce the noise rate. Since the dominant assumption is violated, our method, as well as others, will lose the mapping from the output nodes to the semantic class. However, We believe our method will perform better because there will be some correct similarity labels after the transformation while there is no correct label before the transformation. Actually, this case is **beyond** the consideration of label noise learning. It is of **complementary label learning**, where a complementary label specifies a class that a pattern does not belong to [1]. In this perspective, the task is learnable.
>
> In the literally worst case, where there is no label at all, the task is still kind of learnable by using the unsupervised learning methods. For example, k-means exploits the similarities between unlabeled instances to perform clustering. Our method has a similar principle. Namely, utlizing the similarity supervision to reduce the side-effect of label noise.
>
> [1] Ishida, Takashi, et al. "Learning from complementary labels." *Advances in neural information processing systems*. 2017.
>
>
>
> **Q2:** From algorithm 1 it seems the method requires two independent trainings, shouldn’t that at least double the computational complexity?
>
> **A2:** Compared with learning from clean data, estimating the noise transition matrix (noise rate) do increase the computational cost. However, this step is **necessary** in most label noise learning algorithms [2] [3]. In practice, it **does not double** the computational complexity. For cifar100, we train the model 200 epochs in stage 2 while we only train the model 20 epochs to estimate the noise rate.
>
> [2] Patrini, Giorgio, et al. "Making deep neural networks robust to label noise: A loss correction approach." *CVPR*. 2017.
>
> [3] Xia, Xiaobo, et al. "Are Anchor Points Really Indispensable in Label-Noise Learning?." *NIPS*. 2019.
>
>
>
> **Q3:** Learning with pairs could potentially take longer to converge due to the quadratic increase in the number of input data. How is the claim in section 3.3 that “Class2Simi increases the computation cost *slightly*” supported?
>
> **A3:** To make this clear, suppose we have a training batch containing 3 examples {$\{(x_1, y_1),(x_2, y_2),(x_3, y_3)\}$}. In the standard pipeline, we first forward the model and get {$\hat{y}_1=model(x_1),\hat{y}_2 = model(x_2),\hat{y}_3 = model(x_3)$}, and then compute the $loss = l(\hat{y}_1,\hat{y}_2,\hat{y}_3;y_1,y_2,y_3)$, and then calculate the gradient, and finally backward the model.
>
> In our pipeline, we first forward the model and get {$\hat{y}_1=model(x_1),\hat{y}_2 = model(x_2),\hat{y}_3 = model(x_3)$},
> and then the pairwise enumeration layer calculates the inner products of  3*3 pairs between every two instances.
>
> Next we compute the $loss = l(S_{11}, S_{12}, \cdots , S_{32}, S_{33};H_{11}, H_{12}, \cdots , H_{32}, H_{33})$, where $H$ is generated from {$\{y_1,y_2,y_3\}$}.
> Then we calculate the gradient, and finally backward the model.
>
> Therefore input data **does not quadratic increase** and the additional computation on generating similarity labels and calculating the inner products between every two instances in the pairwise enumeration layer is **time-efficient**. As we cited in the paper, the addition pairwise enumeration layer was proposed in [4]. Quoted from [4], *``**Our empirical finding is that this enumeration introduces a negligible overhead to the training time'**'* . Based on my own experience of conducting experiments, Class2Simi increases the computation cost slightly.
>
> [4] Hsu, Yen-Chang, et al. "Multi-class classification without multi-class labels." *ICLR*. 2019.

---

> > ### Author Response · Authors · 2020-11-16
> > **Thanks for the valuable feedback. We will carefully address the concerns one by one.**
> >
> > **Q4:** I have a concern that the relevant risk to be studied here should be the original category classification risk as opposed to the similarity risk.
> >
> > **A4:** I agree that the class risk is more powerful than similarity risk in this paper. Generalization error is the expected risk of the learned classifier, which is usually bounded by the empirical risk. However, the empirical risk is relate to the objective function, and in this paper, the objective function is of similarity form. Thus we can only derive the similarity risk. Due to the relationships we derived in section 3.2, similarity risk, although is not straightforward as class risk, can justifies why the proposed method works well.
> >
> > **Q5:** Regarding the experiments on Clothing1M and on asymmetric noise ......
> >
> > **A5:** We thank the useful suggestion. However, since Class2Simi is a new perspective of label noise learning, our aim is to justify the effectiveness of using Class2Simi to handle label noise. Thus we only conducted proof-of-concept experiments. The comparison is still fair and the effectiveness of using Class2Simi to handle label noise is still sufficiently justified.
> >
> > **Q6:** Is the same set of hyperparameters used for all the baselines as well as the variants of the proposed approach?
> >
> > **A6:** For fair comparison, we use the same optimization method for all the methods.

---

> > ### Comment · AnonReviewer2 · 2020-11-17
> > **Appreciate the authors effort to address my concerns in detail but the concerns stand**
> >
> > **Q1.** The answer is misrepresenting or missing the point and focuses on the extreme case; -- the point being that reduction of the noise rate for similarity can be (and likely in the general case is) irrelevant here since it can become arbitrarily low without additional class information which is the ultimate goal. It should also be mentioned that the arguments on the extreme case are also inaccurate but that is not central to the discussion so I will not attend to it.
> >
> > **Q3.** Again the point is missed here. The answer is about the computational complexity of a forward/backward pass which clearly will only slightly increase as a result of the new layers. The question was on the convergence time of the new algorithm compared to the standard training. The number of sample pairs (used in the new algorithm), as far as the reviewer understands, is quadratic to the number samples (used in the standard training). This can potentially increase the convergence time, i.e., the number of epochs it takes for the model to converge and not the time it takes to process a batch/update the weights.
> >
> > **Q4.** The answer is unfortunately missing the point again: the ultimate objective function is learning classification under noisy labels. The proposed proxy objective function is learning semantic similarity. The theory shows some properties of the proxy objective. As far as the reviewer understands, without a formal connection between the proxy objective and the ultimate objective, the theory does not provide evidence for the task at hand: learning classification under noisy labels.
> >
> > **Q6.** The question does not concern the optimization method but rather how the hyperparameters are set for each baseline.
> >
> > All in all, despite the rebuttal, the theoretical and empirical concerns are still major and outstanding. Furthermore, the paper has several issues regarding clarity, use of notation, etc. So, I stand by my original "clear reject" rating.

---

> > > ### Author Response · Authors · 2020-11-18
> > > **Regarding the hyperparameter**
> > >
> > > The baselines we have used are of two categories. Methods in one category include the popular transition matrix based methods, i.e., Reweight, Forward, Revision. We internally compare them with the proposed method, i.e., with Reweight & Class2Simi, Forward & Class2Simi, and Revision & Class2Simi. For the second category, we include Co-teaching, APL, and S2E, the latter two are recently published in ICML 2020.
> > >
> > > For all the baseline methods, **we used the same hyperparameters as in the original papers** except for one parameter to locate the anchor points. Anchor points are used to estimate the transition matrix. In the Forward paper, they set the data points with the 97% largest noisy class posteriors as anchor points on MNIST and CIFAR10 and set the data points with the 100% largest noisy class posteriors as anchor points on CIFAR100. Theoretically, they are the ones with the largest clean class posteriors and the largest noisy class posteriors with mild conditions. We have set the data points with the 100% largest noisy class posteriors as anchor points on all datasets. Note that this setting does not affect the effectiveness of the experiments.

---

> ### Author Response · Authors · 2020-11-18
> **On the computational complexity**
>
> We are sorry for missing your points, but now we are a bit confused. For example, what do you mean by "computational complexity"?
>
> Let us assume that the training data are already sampled and fixed; assume also that the model has been initialized. Then the algorithm is still a **randomized** algorithm where the source of randomness is the data shuffling operations inside the stochastic optimizer. All randomized algorithms can be divided into two types: a **Monte Carlo algorithm** has some fixed worst-case time complexity to stop but the result may be incorrect with a small probability; a **Las Vegas algorithm** always leads to the correct result but its worse-case time complexity becomes random or even unbounded. To the best of our knowledge, there is no poly-time Las Vegas algorithm for non-convex optimizations in general, and all popular SGD-like solvers are Monte Carlo algorithms. It implies that **as long as the underlying stochastic solver is the same one, the asymptotic time complexities will not change**, where the complexity means the worst-case analysis by definition.
>
> Empirically speaking, the time needed for convergence of training extremely depends on **the underlying optimizer and the learning rate schedule**, and thus it is hard to compare how many epochs are necessary before convergence... If we decay the learning rate stronger, the convergence will be slower but the converged solution will have better quality as well. Note that **faster convergence to a local minimum cannot imply better training, and better training cannot imply better generalization**. There is an ICML 2020 paper entitled "Do we need zero training loss after achieving zero training error?" showing that by some trick to make training not converged, the model being trained will random walk to regions around flatter minima with higher probability and stay at regions around sharper minima with smaller probability, and thus the trick improves the generalization in the end. Moreover, if we early stop training when it seems converged, we will lose the possibility of enjoying **double descent** test errors/losses. Double descent is one of the hottest topic in deep learning both theoretically and empirically.
>
> If you didn't mean the number of epochs but the computational time of an epoch, we have already answered this question. Note that the number of data to go over is not quadratic: in order to compute a function of pairs for all the data in a mini-batch, we first **forward all the data points**, second pair the high-level hidden representations of points into pairs, third decompose the back-propagated loss signals of pairs into points, and last **backward all the pointwise loss signals**. Therefore, **most computation is still pointwise, and only the computation of the loss part is pairwise**. Since the number of epochs and the number of mini-batches can be regarded as fixed, we compare the averaged computation time of each mini-batch and **our conclusion on the point of the computation time indeed follows an ICLR 2020 paper**.
>
> We are sorry again for missing your points, but what is exactly your points here? Theoretically speaking, it is known in algorithm theory that talking about the time complexity of a Monte Carlo algorithm is nonsense because the time complexity is fixed in order to not affect the probability of correctness. Actually, stochastic optimizer for convex optimizations belongs to Las Vegas, while stochastic optimizer for general non-convex optimizations belongs to Monte Carlo --- did you miss this point? Empirically speaking, is there anything unclear about a single forward+backward pass, such as which part is pointwise computation and which part is pairwise computation?

---

> > ### Author Response · Authors · 2020-11-18
> > **Copy of the post-rebuttal review**
> >
> > PS, the original reply to our rebuttal is "Readers: Paper1431 Authors, Paper1431 AnonReviewer2". There is nothing needed to hide, and we would like to open it for the purpose of an open review.
> >
> > Q1. The answer is misrepresenting or missing the point and focuses on the extreme case; -- the point being that reduction of the noise rate for similarity can be (and likely in the general case is) irrelevant here since it can become arbitrarily low without additional class information which is the ultimate goal. It should also be mentioned that the arguments on the extreme case are also inaccurate but that is not central to the discussion so I will not attend to it.
> >
> > Q3. Again the point is missed here. The answer is about the computational complexity of a forward/backward pass which clearly will only slightly increase as a result of the new layers. The question was on the convergence time of the new algorithm compared to the standard training. The number of sample pairs (used in the new algorithm), as far as the reviewer understands, is quadratic to the number samples (used in the standard training). This can potentially increase the convergence time, i.e., the number of epochs it takes for the model to converge and not the time it takes to process a batch/update the weights.
> >
> > Q4. The answer is unfortunately missing the point again: the ultimate objective function is learning classification under noisy labels. The proposed proxy objective function is learning semantic similarity. The theory shows some properties of the proxy objective. As far as the reviewer understands, without a formal connection between the proxy objective and the ultimate objective, the theory does not provide evidence for the task at hand: learning classification under noisy labels.
> >
> > Q6. The question does not concern the optimization method but rather how the hyperparameters are set for each baseline.
> >
> > All in all, despite the rebuttal, the theoretical and empirical concerns are still major and outstanding. Furthermore, the paper has several issues regarding clarity, use of notation, etc. So, I stand by my original "clear reject" rating.

---

> > ### Comment · AnonReviewer2 · 2020-11-18
> > **the other points**
> >
> > Thank you for the answer on the computational complexity, however, please read the original review and my reply again and carefully: I pointed at the *computational complexity* in the context of two independent trainings (which you answered in your original rebuttal) and *convergence time* in the context of the empirical loss being now defined on pairs as opposed to singular samples. This can increase the "convergence time" to a similar error rate as for instance data augmentation does. I find your current reply on explaining "computational complexity" of Monte Carlo methods impertinent to my question.
> >
> > In any case, please answer all the raised points above before I finalize my review.
> >
> > ---------------------------------
> >
> > I am ignoring the non-scientific comments and the insinuation (on hiding my review), I suggest the authors focus on addressing the points to not waste the time and energy of themselves as well as the reviewers. Please do not reply to this point (take it or leave it). Thank you.

---

> > ### Author Response · Authors · 2020-11-18
> > **It seems that we need the area chair to step in again**
> >
> > It seems that we need the area chair to step in again. We have answered that "faster convergence to a local minimum cannot imply better training, and better training cannot imply better generalization", but the reviewer's point is that data augmentations make the convergence of training slower so that they are bad?
> >
> > Moreover, the reviewer said again and again like "misrepresenting or missing the point", "again the point is missed", "unfortunately missing the point again", "please read the original review and my reply again and carefully", and "please do not reply to this point (take it or leave it)". Who distracts the discussion to be non-scientific in this thread?

---

> > ### Comment · Area_Chair1 · 2020-11-18
> > **Discussion**
> >
> > Folks, again, let's please keep this discussion focussed on technical questions. Other points are not constructive.
> >
> > My understanding is that compared to standard learning, the proposed algorithm conceptually involves training on a quadratic number of samples. A natural question is thus, how much of an overhead (if at all) does this pose? More specifically, one may consider:
> >
> > (1) the cost of individual updates on a minibatch
> >
> > (2) the number of steps needed to reach a fixed training error/loss
> >
> > (3) the number of steps needed to reach a fixed test error/loss
> >
> > I gather that (1) has been adequately resolved in the original response. As I understand, R2 is unsure about (3). I understand the comment about data augmentation to mean that augmentation similarly increases the number of samples, and requires more steps to reach a fixed test error.
> >
> > R2, is that a fair summary? What are your thoughts on the authors' remarks on the potential mismatch between optimization and generalization?

---

> > > ### Comment · AnonReviewer2 · 2020-11-18
> > > **Re: Discussion**
> > >
> > > Correct summary. Thank you.
> > >
> > > As stated in my reply, I do not have any comment on the remarks since I find them orthogonal to my question.
> > >
> > > In my proposed analogy: I do not think one can, at least in the general case, simply claim that: "~~Class2Simi~~ [data augmentation] increases the computation cost slightly"
> > >
> > > Apart from this, the other concerns regarding the theories and clarity are still unaddressed. The responses to concerns regarding the experimental setup are unconvincing.

---

> > > > ### Comment · Area_Chair1 · 2020-11-19
> > > > **Re: Discussion**
> > > >
> > > > Thanks R2!
> > > >
> > > > I'd like to clarify further re: the point about convergence time. I understand that the reported experiments employ a fixed number of epochs for all methods. Assuming the compute time per epoch/minibatch update is similar to the baseline (per point (1)), it would suggest that at least on the reported benchmarks, the time to reach a well-generalizing solution is comparable.
> > > >
> > > > Does the above sound right to you? If so, is the concern essentially that for other, more complex tasks, one may require more epochs with the proposed method? Alternately, if this misses the crux of your concern, please let me know. Thanks!

---

> > > > ### Author Response · Authors · 2020-11-19
> > > > **On hyperparameter optimization**
> > > >
> > > > We have already addressed the hyperparameter-tuning issues but R2 has ignored it. Let me address it once more in a different manner.
> > > >
> > > > a. Is the same set of hyperparameters used for all the baselines as well as the variants of the proposed approach?
> > > >
> > > > *No.*
> > > >
> > > > b. How are the hyperparameters optimizations done?
> > > >
> > > > *In the current paper, it's done by reusing the same hyperparameters for each baseline method used by the original authors in the original papers. In the original papers, we don't know how it was done.*
> > > >
> > > > c. In particular, which variant of the method or baselines are the hyperparameters optimized on?
> > > >
> > > > *Assuming the original authors well tune the hyperparameters for their own methods, the hyperparameter optimization can be regarded as done on each (baseline and proposed) method separately.*
> > > >
> > > > d. In our experience, when it comes to noisy labels, it is quite common that different methods perform better with different sets of hyperparameters, so it's important to optimize the parameters per method.
> > > >
> > > > *Yes, the hyperparameters can indeed be regarded as optimized per method.*
> > > >
> > > > Anything still unclear?

---

> > > > > ### Comment · Area_Chair1 · 2020-11-19
> > > > > **Re: On hyperparameter optimization**
> > > > >
> > > > > Thanks for the further comments on this point. However, one more time, please refrain from asides such as "R2 has ignored it". These are not conducive to a substantive discussion.
> > > > >
> > > > > R2, do you have further thoughts on the hyperparameter issue?

---

> ### Comment · Area_Chair1 · 2020-11-18
> **Discussion on noise rate for similarity labels**
>
> I'd like to follow up on the point R2 raises regarding the noise rate for similarity labels.
>
> My understanding of R2's concern is whether low noise rate on similarity labels implies learnability with respect to the clean labels. They offered as an example a situation where (as I understand) all labels are chosen uniformly at random, and thus contain no information about the true labels. The concern is that in such cases, one may have low noise rate on dissimilar labels. If this is so, this would be an extreme example of a case where low noise rate on dissimilar labels does not imply learnability with respect to the clean labels.
>
> R2, is that a fair summary of your concern?

---

> > ### Author Response · Authors · 2020-11-18
> > **Clarification?**
> >
> > Dear AC, I would like to ask for clarification: is the concern whether low noise rate on "similarity" or "dissimilarity" labels implies learnability? Your first sentence says similarity but last sentence says dissimilarity.

---

> > > ### Comment · Area_Chair1 · 2020-11-18
> > > **Re: Clarification?**
> > >
> > > Sorry for the confusion. My understanding is that the concern is whether low noise on *dis*similar labels implies learnability.
> > >
> > > R2, please correct me if I misunderstood.

---

> > ### Author Response · Authors · 2020-11-19
> > **On noise rate for dissimilarity labels**
> >
> > OK, let us temporarily admit that the noise rate of negative pairs **can be very low** given fully random labels. Assume also that the noise rate of positive pairs **cannot be very high**, in order to keep the overall noise rate sufficiently low (we will overturn this assumption later). Then, this is an issue of **classification calibration**.
> >
> > We distinguish two scenarios: in the **practical case**, the causality is **class label-->pair label**; in the **theoretical case**, the causality is **pair label-->class label**. Considering the theoretical case, if a surrogate loss is classification-calibrated, minimizing it will leads to minimizing the zero-one loss on the class-label random variable in the limit case, and let us call it **learnable** with a bit abuse of terminology. If a loss is not classification-calibrated, we cannot guarantee the worse-case learnability, which **cannot imply average-case non-learnability either**.
> >
> > Note that this topic studies the limit behaviors of surrogate losses given uncountably infinite data, and the theoretical results are called **excess risk bounds** involving much more complicated theory than **estimation/generalization error bounds**. There is no guarantee that a classification-calibrated loss working well in the limit case must also work well in the finite-sample case. The latter needs estimation/generalization error bounds. What we proved is an estimation/generalization error bound guaranteeing the **weak consistency of training**, and often this is already the best we can offer in terms of the worst-case error analysis in weakly supervised learning. To the best of our knowledge, previous label-noise papers also provide such bounds rather than excess risk bounds, and very few papers studied the classification calibration (there was 1 ICML 2020 paper on binary classification under label noise).
> >
> > It seems fine to use non-classification-calibrated losses in practice. According to [1], the multi-class margin loss (i.e., 1-vs-rest loss) and the pairwise comparison loss (i.e., 1-vs-1 loss) are **proved to be non-calibrated**, but they are still the main multi-class losses in [2,3]. For semi-supervised learning and adversarial learning, it is even proved that **all margin-based or all convex losses are non-calibrated** [4,5]. However, people still prefer using margin-based and/or convex losses for training models in semi-supervised learning and adversarial learning. The classification calibration is just one theoretical property and not the most important one. The genuine desideratum is **a good loss that works in the practical case (finite-sample, class-->pair causality, and average-case) rather than the theoretical case (asymptotic, pair-->class causality, and worst-case)**.
> >
> > BTW, a key difference is that the above losses are proved to be non-calibrated while our loss is not proved to be calibrated. Hope the last paragraph will not mislead the reviewers.
> >
> > Last but not least, let us focus on the extreme case and look at the assumption. For class-balanced data sets with 10 classes, when the labels are random, **the noise rate of positive pairs is 0.9 and the noise rate of negative pairs is 0.1**. Pairwise learning is a special case of binary classification, and a noise rate of 0.9 is too high to learn. Even if we don't consider the noise rate of positive pairs because they are the minority class, in label-noise learning a noise rate of 0.1 is already noticeable and should be handled by the algorithm.
> >
> > In fact, an incorrect positive label is much more adversarial even though they are the minority class, quoted from our reply: *"In practice, it is common that one class have more than one clusters, while it is rare that two or more classes are in the same cluster. If there is a flip as similar-->dissimilar and based on it we split a (latent) cluster into two (latent) clusters, we still have some high chance to label these two clusters correctly later. If there is a flip dissimilar-->similar and based on it we join two clusters belonging to two classes into a single cluster, we nearly have zero change to label this cluster correctly later. As a consequence, the latter form is more adversarial, and without any special design of corrections, the standard pairwise training is hopeless to handle this form of label noise."*
> >
> > [1] Ambuj Tewari and Peter L. Bartlett, On the Consistency of Multiclass Classification Methods, JMLR 2007.
> >
> > [2] Mehryar Mohri, Afshin Rostamizadeh, Ameet Talwalkar. Foundations of Machine Learning, The MIT Press 2018.
> >
> > [3] Shai Ben-David and Shai Shalev-Shwartz. Understanding Machine Learning: From Theory to Algorithms, Cambridge University Press 2014.
> >
> > [4] Jesse H. Krijthe, Marco Loog. The Pessimistic Limits and Possibilities of Margin-based Losses in Semi-supervised Learning, NeurIPS 2018.
> >
> > [5] Han Bao, Clayton Scott, Masashi Sugiyama. Calibrated Surrogate Losses for Adversarially Robust Classification, COLT 2020.

---

### Official Review · AnonReviewer5 · 2020-11-07
**Interesting idea but the analysis is problematic**

**Rating:** 3
**Confidence:** 4

**Review:**

This paper proposes a new algorithm on learning noisy datasets by transforming class labels into pairwise similarity labels. It also gives some theoretical analysis on the fact that the induced similarity noise transition matrix works better than the class noise transition matrix. The paper empirically demonstrates that the proposed method works well on several synthetic datasets and a large-scale real-world dataset.

Strengths:
- I believe the idea to use pairwise similarity as supervision is novel and interesting, and it is easy to implement.

Weaknesses & Questions:
- I think the analysis is a bit problematic. Th. 2 shows that when the number of classes is large (>8), the noise rate of similarity labels is less than class labels. And the authors use Th. 3 to prove that if the noise rate of transition matrix decreases the model will have a better generalization. However, as far as I understand, the supervision effect of the pairwise label differs a lot between positive and negative labels. In fact negative pairwise supervision is not very meaningful as there are a lot of gradient directions that can minimize the loss. Thus I think evaluate on the noise ratio of the whole pairwise similarity matrix is not very meaningful. And since the supervision effect of class-level and similarity-level labels is so different, it casts questions on the whole theoretical analysis in my understanding.
- The baselines on CIFAR seem too low compared with the SOTAs, e.g. [1], and the improvement of the proposed method is limited. And the final result is not comparable as well. For example, under the setting of 0.5 sym noise of CIFAR-10, the best result of the proposed method is 81.15, while [1] has 84.78.

[1] Chen, Pengfei, et al. "Understanding and Utilizing Deep Neural Networks Trained with Noisy Labels." International Conference on Machine Learning. 2019.

-----------------------------
Post Rebuttal Modification

Regarding A1: I agree with R2 that the theory has major concerns and the authors were not able to fix it during rebuttal. I think we need to be clear that whether the method can work empirically and whether the provided theory can explain it are two problems. Now it seems to me that it is clear that the theory is wrong, and the problem is that the authors did not take into account the difference of the class-wise labels and pairwise labels. I suggest the authors to change the theory completely or remove the theory before next submission.

Regarding A2: I don't chase SOTAs and I could certainly appreciate works that give nice theoretical insight but limited improvement. Now that the theory is wrong I have to be critical about the experiments. Since the performance is much worse than STOA, it is no longer clear whether the proposed algorithm works or it's just because the baselines are too bad.

I adjusted my rating from 5 to 1.

-----------------------------
Regarding the authors' 2rd and 3rd responses

First, please allow me to clarify that my wording "the theory is wrong" means the theoretical justification on why the proposed algorithm can benefit from the transformation and achieve better performances is wrong, as the authors wrote “This theoretically justifies why the proposed method works well” in their submission. The major flaw/concern has been raised by R1(Q1) and myself(Q1), and the authors’ responses on these two questions are not convincing. This is the concern that I have been asking, so I assume it is not “vague”. I don’t see any potential way to fix this major concern in the current theoretical justification sketch, so I think this submission needs a major revision. I want to give my apology if my wording "the theory is wrong" leads to misunderstanding to the authors or other reviewers.

Second, I would like to see ACs or PCs to step in and let me know if I could rate the submission as 1 in this case. I have temporarily increased my rating from 1 to 3 as it has been questioned by the authors, especially the author who “have served as a reviewer 100+ times and as an area chair 10+ times for top conferences like NeurIPS/ICML/ICLR”.

What’s more, I would also like to request apologies from the authors. The wording “angry” is unpleasant and misleading. As the author asked, “what are you angry for?”, I’m not angry at all. I simply adjusted my post-rebuttal rating with my expertise after reading the authors’ responses and other reviewers’ comments.

Finally, I would like to remind the author who “have served as a reviewer 100+ times and as an area chair 10+ times for top conferences like NeurIPS/ICML/ICLR”, one of the main rules of academic writing is to avoid using second person. I hope this will be helpful.

Best

---

> ### Author Response · Authors · 2020-11-16
> **Thanks for the valuable feedback.  We will carefully address the concerns one by one.**
>
> **Q1:** I think the analysis is a bit problematic.
>
> **A1:** We agree that the supervision effect of class-level and similarity-level labels is so different. Class supervision can give a explicit optimization direction. As you mentioned, there are a lot of gradient directions that can minimize the loss for negative pairwise supervision, and it also applies to the positive pairwise supervision. However, when the similarity pairwise supervision is sufficient, the gradient directions can be determined. For example, assume we have three examples of three classes, i.e., ${\{(x_1, 1),(x_2, 2),(x_3, 3)\}}$, and then we generate three example-pairs ${\{(x_1, x_2),(x_2, x_3),(x_3, x_1)\}}$. They are all negative pairs, but the gradient direction is determined for them due to the constraints between every two instances.
>
> In any case,  although the negative pairwise supervision is less informative than the positive pairwise one, and pairwise supervision is less informative than the class supervision, the amount of negative pairwise supervision is $k$ times as much as that of positive pairwise supervision, and the amount of pairwise supervision is $k^2$ times as much as that of class supervision, where $k$ is the number of classes.
>
> Therefore, analyzing the noise rate of the whole similarity labels is meaningful.
>
> **Q2:** The baselines on CIFAR seem too low compared with the SOTAs, and the improvement of the proposed method is limited. And the final result is not comparable as well.
>
> **A2:** Thanks for pointing out this. Since Class2Simi is a new perspective of label noise learning, our aim is to justify the effectiveness of using Class2Simi to handle label noise. Thus we only conducted proof-of-concept experiments. Achieving the state-of-the-art accuracy is not our purpose now but would be considered in the future work.

---

> ### Author Response · Authors · 2020-11-17
> **We are very confused about the comments**
>
> The reviewer concluded that “it is clear that the theory is wrong”. We double checked the proof and could not find a single line of the proof which is wrong. We will appreciate it if the reviewer could point out which part is wrong. As serious researchers, we cannot accept this judgment without any evidence. We feel such a comment is not ethically acceptable if there is no evidence.

---

> ### Author Response · Authors · 2020-11-17
> **On "it is clear that the theory is wrong"**
>
> Dear R5,
>
> It seems that you are fairly **angry with** our submission/rebuttal given your vague post-rebuttal review and your action of adjusting the rating from 5 to 1, but what are you **angry for**?
>
> I need to explain the meaning of **wrong** in "1: Trivial or wrong". Technically speaking, there must be some **fatal** error: for algorithm paper, the fatal error should be in the derivation of the proposed algorithm; for theory paper, the fatal error should be in the proof of lemmas or theorems, excluding the corollaries. The existence of the fatal error must be so bad that **the major contributions become useless** and then **the main conclusion and take-home message are overturned**. Note that fixable proof errors in theory papers are not regarded as wrong. On the other hand, even fatal proof errors in theorems in algorithm papers are only regarded as wrong, if they are in the motivation part and affect the correctness of the proposal; they are still regarded as **fixable**, if they are in the justification part and cannot go into the algorithm design.
>
> Thus, can you be more specific in **which equality or inequality does not hold**? We are happy to revise the paper to fix any fixable proof error.
>
> However, if your judgment "it is clear that the theory is wrong" did not mean fatal errors in our proofs, but meant the implications of our theorems are not meaningful, I must say that you have **misunderstood** the meaning of wrong in "1: Trivial or wrong" and please reconsider your rating. Whether it is wrong is an **objective** matter and whether it is meaningful is a **subjective** matter. If you personally think our theorems are not meaningful, you cannot judge our theory as wrong and adjust the rating from 5 to 1. I have served as a reviewer 100+ times and as an area chair 10+ times for top conferences like NeurIPS/ICML/ICLR, so I am quite sure my understanding of wrong in "1: Trivial or wrong" is correct.
>
> As I said, we are happy to revise the paper to fix any fixable proof error. We are looking forward to your help for finding them. In the worst case that you cannot find any fatal error in our theorems but insist "it is clear that the theory is wrong" and rate our work "1: Trivial or wrong", your action will raise an **ethical** issue that needs to be handled by the area chair or even the program co-chairs.
>
> Thanks!
>
> Authors

---

> ### Author Response · Authors · 2020-11-17
> **On the last modification**
>
> Now, I would like to see the program chairs to step in too! This is because R5 clearly **either was biased or misunderstood the meaning of rating**.
>
> About R5's two actions, there are two possibilities: if s/he originally well understood the meaning of "1: Trivial or wrong" and knew it is not same as her/his minded **applicability** issue of our theorems, this on-purpose rating as 1 is already an **ethical** issue as a reviewer when a reviewer judges **the value of correct contributions** according to the **personal taste**. Rating as 3 seems not ethical even though still judging the value of contributions by the personal taste, but actually this is more serious since the bias is less obvious! Without my reply, this "1: Trivial or wrong" will not be changed to "3: Clear rejection", so how about the authors who are all junior researchers and cannot point this issue out? It would be extremely unfair to the authors!
>
> On the other hand, if R5 indeed misunderstood the meaning of "1: Trivial or wrong", this is certainly not an ethical issue. However, this also implies that R5 is **not so familiar with this scale of the review form**... According to my personal and only personal experience, R5 should be **an expert from another area** such as deep learning theory. I am a bit confused why such an expert could be not so familiar with the scale of the review form.
>
> Other evidences as follows. R5 modified her/his post two times, **not for typos but for critically new information**. This implies that R5 is not familiar with the ICLR reviewing system. When a post owner modifies the original post, no one can receive email notification... R2 who also gave us 3 seems much more experienced in this reviewing system: R2 at least knows how to post new information and draw authors' attention for continuing the discussion.
>
> Furthermore, R5 cannot really distinguish **scientific writing** and **academic writing**. In scientific writing, we need to in principle avoid to use the first person (math paper is the only exception). In academic writing especially for arguments, it is free to use the first and second person. Following your reply, please do not **remind others about academic writing** because it is also "unpleasant and misleading".

---

> ### Comment · Area_Chair1 · 2020-11-18
> **Discussion**
>
> Folks, I'd like to suggest we lower the temperature of this discussion, and focus on the core scientific questions.
>
> *Authors*: It is perfectly legitimate to seek clarification on the justification behind a change of score, if it is unclear. However, in doing so, I suggest we assume good intent on the part of all reviewers (who have provided detailed comments on the paper), and not try to offer non-scientific reasons behind this change.
>
> *R5*: in the interest of steering the discussion towards the scientific issues, could I request you to reply to this with a summary of whether you judge that the central technical result of the paper is:
> (a) mathematically incorrect, with no apparent fix; or
> (b) mathematically correct, but does not actually show what is claimed
>
> If (a), could you specify which particular theorem statements, equations or arguments are incorrect? If (b), could you summarize what is claimed in the text, versus what is shown in the theorem(s)?
>
> Thanks.

---

> > ### Author Response · Authors · 2020-11-18
> > **We will stop doing so by your suggestion!**
> >
> > We are sorry for seeking the non-scientific reasons behind this change, because we have never seen a decrease of 4 (5-->1 or 10-->6) without the identification of a truly fatal error among the papers we have submitted/reviewed/chaired (let alone 5-->1-->3). Note that our paper is not a theory paper, and the theory is for explaining why transforming noisy class labels to noisy similarity labels can be a good idea. Even if the theorems are mathematically wrong, they will not affect the correctness of the proposed algorithm which is the major contribution of the paper. We will stop doing so by your suggestion!

---

> > ### Comment · AnonReviewer5 · 2020-11-18
> > **discussion towards the scientific issues**
> >
> > I would like to thank AC for starting this thread. I judge that the central technical result of the paper is (b) mathematically correct, but does not actually show what is claimed.
> >
> > Specifically, after introducing Th. 1,2,3, the authors claim that “This theoretically justifies why the proposed method works well”, so I assume there exists a theoretical analysis, with reasonable assumptions and simplifications, can show that the proposed method can benefit from the transformation and achieve better performance. However, the theorems that the authors introduced are mainly: [Th2], which shows that pairwise supervision has less ratio of noise compared with class-wise supervision, and [Th3], which studies how the error in similarity matrix affects generalization error on similarity risk. I can hardly come to the conclusion given the above theorems. The major technical flaw I found is that a lower matrix-wise error does not necessarily mean a better classification model. For a balanced large set with sufficient number of classes, a similarity model that only predicts 0(dissimilar) can achieve a risk near to 0. On the contrary, its classification risk will be large. The intuition after this, as I have stressed, is the difference between positive and negative pairs. In the rebuttal, the authors first agree with my point above. And then argue that “In any case, although the negative pairwise supervision is less informative than the positive pairwise one, and pairwise supervision is less informative than the class supervision, the amount of negative pairwise supervision is k times as much as that of positive pairwise supervision, and the amount of pairwise supervision is k^2 times as much as that of class supervision, where k is the number of classes.” This argument is not valid as well, as the authors must provide evidence that the supervision effect of positive pair is at most k^2 times larger than negative pairs.
> >
> > I’m not very certain that my evaluation is correct in the original review, so I rate my confidence as 3 and rate the submission as 5. I have much more confidence after reading other reviewers’ comments as well as the rebuttal. So I increased my confidence and lowered the score. As a junior reviewer who is serving as a reviewer for ICLR for the first year, I would like to reiterate my apology if my adjustment in score seems not so standard. However, I would like to point out I am doing all I can to give this submission a fair recommendation.
> >
> > Since the authors further emphasized that their submission is not a theory paper, I would also like to reiterate that my final recommendation also takes the empirical performance into consideration. I’m generous on whether a paper needs to beat SOTAs if the paper delivers interesting theories or insights. However, if the authors want to only emphasize on the proposed algorithm, my concern is that the performance is far lower than a ICML19[1] paper, and due to this issue, I would still vote for rejection.

---

> > ### Author Response · Authors · 2020-11-18
> > **Thanks for the clarification!**
> >
> > Thanks for the clarification! We address your concerns step by step.
> >
> > **Step 1** Consider standard pointwise binary classification where two classes are imbalanced. Since deep networks can memorize any fixed training data and achieve zero training error or even (almost) zero training loss, the phenomenon that the model predicts all training data to the majority class cannot happen in practice. It may only happen for underfitted models with insufficient capacity (depending on the amount of data and the intrinsic complexity of a single data point itself). Note that deep networks approximate continuous functions, and hence the model will neither predicts all test data to the majority class.
> >
> > **Step 2** We have assumed that the label corruption is not adversarial, namely, the label transition matrix is still diagonally dominant. Before fitting the similarity/dissimilarity labels (constructed on the fly), we have pretrained the model for multi-class classification fitting the noisy class labels. This model serves as a very good initialization for the model fitting noisy pairwise labels. As a result, the aforementioned phenomenon cannot happen in practice.
> >
> > **Step 3** What we have agreed is that class labels and pairwise labels are different...
> >
> > **Step 4** Let us admit that a similarity label and a dissimilarity label are different, more specifically, they have different importance. This does not mean a pairwise label flip similar-->dissimilar and a flip dissimilar-->similar should have different importance. Note that this flip is about the labels given for training at the supervision level, rather than about the model prediction at the training level.
> >
> > **Step 5** Let us also admit that the two types of pairwise label flips have different importance. Then, dissimilar-->similar should be more adversarial and thus more important, explained below. In practice, it is common that one class have more than one clusters, while it is rare that two or more classes are in the same cluster. If there is a flip as similar-->dissimilar and based on it we split a (latent) cluster into two (latent) clusters, we still have some high chance to label these two clusters correctly later. If there is a flip dissimilar-->similar and based on it we join two clusters belonging to two classes into a single cluster, we nearly have zero change to label this cluster correctly later. As a consequence, the latter form is more adversarial, and without any special design of corrections, the standard pairwise training is hopeless to handle this form of label noise.
> >
> > To sum up, the phenomenon that the model predicts all pairwise data to be dissimilar cannot happen in practice, and the label flipping in supervision and that in model prediction during training are completely different matters.
> >
> > BTW, in the research area of label noise, the latter is quite well known that the label flipping in supervision and in model prediction are completely different in cases where the labels are pointwise class labels. We are sorry that we did not explain this point in detail due to limited space.
> >
> > PS, we are still confused about **which pairwise label do you think is more important**? Considering the different numbers of positive and negative pairs, you have asked us to **provide evidence** on **supervision effect**. However, we cannot find the formal definition of supervision effect. Without a clear definition, how can we provide the required evidence? Please show us an authoritative book or paper where we can find the formal definition in mathematics, and then we will consider how to theoretically/empirically show it.

---

> > ### Author Response · Authors · 2020-11-18
> > **On "the authors must provide evidence that the supervision effect of positive pair is at most k^2 times larger than negative pairs"**
> >
> > After careful discussions with many experts, we try to address this concern by defining the supervision effect as the **information gain**.
> >
> > Suppose we have two i.i.d. random variables $Y_1$ and $Y_2$. Without the pairwise label, the entropy is $H(Y_1,Y_2) = H(Y_1) + H(Y_2) = 2H(Y)$.
> >
> > Consider a positive label. We have $H(Y_1,Y_2|Y_2=Y_1) = H(Y_1|Y_2=Y_1) + H(Y_2|Y_1,Y_2=Y_1) = H(Y_1) + 0 = H(Y)$, and the information gain is $H(Y)$.
> >
> > Consider a negative label. We have $H(Y_1,Y_2|Y_2 \neq Y_1) = H(Y_1|Y_2 \neq Y_1) + H(Y_2|Y_1,Y_2 \neq Y_1)$. By the definition of conditional entropy, $H(Y_2|Y_1,Y_2 \neq Y_1) = \sum_{i=1}^k p(Y_1=i) * H(Y_2|Y_1=i,Y_2 \neq Y_1) = \sum_{i=1}^k p(Y_1=i) * H(Y_2|Y_2 \neq i)$. It is assumed that $Y_1$ and $Y_2$ share the same distribution, and $H(Y_2|Y_1,Y_2 \neq Y_1)$ is maximized when the distribution of $Y$ is **uniform**. In the sequel, we focus on the uniform case.
> >
> > In the uniform case, the information gain of seeing a positive label is $\ln(k)$. The information gain of seeing a negative label is at least $\ln(k)-\ln(k-1)$. The ratio of gains is at least $(\ln(k)-\ln(k-1))/\ln(k) = 1-\ln(k-1)/\ln(k)$.
> >
> > For any reasonable value of $k$, it holds that $1-\ln(k-1)/\ln(k) > 1/k^2$, though it cannot hold for all possible $k$. For example, when $k=10$, LHS is 0.045757 and RHS is 0.01; when $k=100$, LHS is 0.002182 and RHS is 0.0001; when $k=10^{10}$, LHS is 4.34297e-12 and RHS is 1e-20.
> >
> > Hence, **we have demonstrated that for at least $k\le10^{10}$, the information gain of seeing a negative label is at least $1/k^2$ times of the information gain of seeing a positive label**.
> >
> > PS, when the reviewer claimed some scientific statements against our submission/rebuttal like this is wrong and that is wrong, could the reviewer please also provide such theoretical or empirical evidence?
> >
> > -----
> >
> > Sorry. We should study the function $k^2*(1-\ln(k-1)/\ln(k))$ and see if it is greater than one. The function takes 4.0, 3.3216, 3.3202 when $k$ is 2, 3, 4. The function is monotonically increasing when $k\ge4$. Thus the claim holds for all $k\ge2$.

---

> > > ### Comment · Area_Chair1 · 2020-11-18
> > > **Discussion on supervision effect**
> > >
> > > I read the reviewer's statement as saying that for your argument to go through, you need to quantify the relative utility of learning from a negative versus positive pair. This seems a reasonable request for clarification.
> > >
> > > The detailed responses on this point are appreciated. R5, do you have any thoughts on this?

---

> > ### Author Response · Authors · 2020-11-19
> > **On "a lower matrix-wise error does not necessarily mean a better classification model"**
> >
> > Yes, the former doesn't **always** mean the latter, but the former **often** means the latter **in practice**. We have addressed this concern in a post entitled "on noise rate for dissimilarity labels". Could you please refer to that post and see if it addresses your concern (as R2 has the same concern)? Thanks!

---

> ### Author Response · Authors · 2020-11-18
> **Regarding the SOTA paper: [1] Chen, Pengfei, et al. "Understanding and Utilizing Deep Neural Networks Trained with Noisy Labels." ICML. 2019.**
>
> The reviewer commented that “it is no longer clear whether the proposed algorithm works or it's just because the baselines are too bad”. We would like to mention that we have a better performance of the same baseline used in [1]. Under the setting mentioned by the reviewer, i.e., 0.5 sym noise of CIFAR-10, the baseline Forward has the accuracy of **76.02** in [1] (see Table 1 therein); while the baseline Forward has a higher accuracy of **78.10** in our paper (see Table 1). Based on this, the internal comparison with Forward, i.e., Forward & Class2Simi outperforms Forward, shows the effectiveness of the proposed method.

---

> > ### Comment · Area_Chair1 · 2020-11-23
> > **Re: Regarding the SOTA paper**
> >
> > **Authors**: could you comment on whether Class2Simi is also applicable on top of the best-performing method of [1]? Thanks.

---

> > > ### Author Response · Authors · 2020-11-24
> > > **Regarding the SOTA paper**
> > >
> > > Under the setting of 0.5 symmetric noise on \textit{CIFAR10}, INCV [1] achieves a classification accuracy of **84.78**, using ResNet-32 (more layers but fewer filters and fewer parameters). We conduct experiments using two other networks with more parameters, i.e., ResNet-18 and ResNet-26. For [1], we use the implementation provided by the author, and all other settings are kept the same. For Class2Simi, we implement it on Revision. The weight decay is set to 5e-4 and all other settings are kept the same. The results of ResNet-32 are quoted from the [1] and our submission, which is over 5 trials. The other results are over 3 trials. Note that both the methods may be overfitted on ResNet-18.
> > >
> > > | 0.5 Symmetric Noise | ResNet-32      | ResNet-18      | ResNet-26      |
> > > | ------------------- | -------------- | -------------- | -------------- |
> > > | INCV [1]            | 84.78$\pm$0.33 | 79.49$\pm$0.23 | 80.84$\pm$0.37 |
> > > | Class2Simi          | 81.15$\pm$0.32 | 82.28$\pm$0.41 | 85.54$\pm$0.17 |
> > >
> > > From the Table we can see that Class2Simi achieves an accuracy of **85.54** with ResNet-26, exceeding the best accuracy in [1]. This is because INCV employs a sample-selection strategy, which may not exploit the non-confident data well, while our method ($T$-based) is classifier-consistent, which exploits all the noisy data. Therefore, when employing a network with more parameters (or stronger expressive power), our method can better exploit the non-confident data and achieve better performance. The detailed information about the number of parameters of the employed networks are presented below:
> > >
> > >
> > > | Total params         | ResNet-32 | ResNet-18  | ResNet-26 |
> > > | -------------------- | --------- | ---------- | --------- |
> > > | Keras ([1])          | 470,602   | 11,192,458 | 2,930,810 |
> > > | Pytorch (Class2Simi) | 464,154   | 11,173,962 | 2,923,162 |
> > >
> > > Since the original implementation of [1] is on Kears while ours is on Pytotch, the exact number of total parameters of the same network on different frameworks is slightly different. Overall, given the same network, there are more parameters on Keras than Pytorch. Note that for INCV, the hyperparameter for selecting confident examples should be the same; while the hyperparameters for optimization could be different and should be optimized (We used the same hyperparameters optimized for ResNet-32).
> > >
> > > Regarding whether the proposed method could be applied on top of the best-performing method of [1], our answer is yes. There are two possible ways at least. One is that, we could employ the sample-selection method as proposed in [1] for the noisy pairwise learning; The other one is that, we can first employ the sample-selection method for selecting confident pointwise labeled examples and then employ the proposed Class2Simi method onto the selected data by transforming them into noisy pairwise labeled data. We assume the latter one could have better performance as the noise is reduced twice.

---

### Public Comment · ~Ehsan_Amid1 · 2020-11-10
**Please consider referencing/comparing to these more recent works**

I would like to point out that our work (Amid et al. 2019a) extends the Generalized CE loss (Zhang and Sabuncu 2018) by introducing two temperatures t1 and t2 which recovers GCE when t1 = q and t2 = 1. Our more recent work, called the bi-tempered loss (Amid et al. 2019b) extends these methods by introducing a proper (unbiased) generalization of the CE loss and is shown to be extremely effective in reducing the effect of noisy examples. Please consider referencing/comparing to these papers.

(Amid et al. 2019a) Amid et al. "Two-temperature logistic regression based on the Tsallis divergence." In The 22nd International Conference on Artificial Intelligence and Statistics (AISTATS), 2019.

(Amid et al. 2019b) Amid et al. "Robust bi-tempered logistic loss based on Bregman divergences." In Advances in Neural Information Processing Systems (NeurIPS), 2019.

---

### Author Response · Authors · 2020-11-19
**On theoretical connection from pairwise classification to pointwise classification**

We found a theoretical preprint on arXiv of 22 pages investigating this connection. The title is *Similarity-based Classification: Connecting Similarity Learning to Binary Classification* and the abstract is quoted below.

*In real-world classification problems, pairwise supervision (i.e., a pair of patterns with a binary label indicating whether they belong to the same class or not) can often be obtained at a lower cost than ordinary class labels. Similarity learning is a general framework to utilize such pairwise supervision to elicit useful representations by inferring the relationship between two data points, which encompasses various important preprocessing tasks such as metric learning, kernel learning, graph embedding, and contrastive representation learning. **Although elicited representations are expected to perform well in downstream tasks such as classification, little theoretical insight has been given in the literature so far.** In this paper, we reveal that a specific formulation of similarity learning is strongly related to the objective of binary classification, which spurs us to learn a binary classifier without ordinary class labels---by fitting the product of real-valued prediction functions of pairwise patterns to their similarity. Our formulation of similarity learning does not only generalize many existing ones, but also admits an excess risk bound showing an explicit connection to classification. Finally, we empirically demonstrate the practical usefulness of the proposed method on benchmark datasets.*

The theoretical results in this preprint guarantee that **when the pairwise labels are all correct, for the special case $k=2$, a good model for predicting similar/dissimilar pairs must also be a good model for predicting the original classes**, under mild assumptions. It's really non-trivial to extend it to the general case $k>2$ because their proof technique is quite specially designed for their purpose. Nevertheless, **it significantly increases the chance of being classification-calibrated for [1]**. If a good model for predicting similar/dissimilar pairs must also be a good model for predicting the original classes holds for [1], it will also hold for our submission **given our current mathematically correct theorems**.

[1] Yen-Chang Hsu, Zhaoyang Lv, Joel Schlosser, Phillip Odom, Zsolt Kira. Multi-class Classification without Multi-class Labels, ICLR 2019.

-----

We also found a NeurIPS 2020 paper (accepted but not published yet) [2] that studies *"learning from aggregate observations where supervision signals are given to sets of instances instead of individual instances, while the goal is still to predict labels of unseen individuals"*. [2] has established the connection from **not only pairwise similarity but also triplet comparison** to multi-class classification, under not mild but relatively strong distributional assumptions. At least, [2] proves the possibility that a good model for predicting similar/dissimilar pairs must also be a good model for predicting the original classes **considering multi-class classification**.

[2] Yivan Zhang, Nontawat Charoenphakdee, Zhenguo Wu, Masashi Sugiyama. Learning from Aggregate Observations, NeurIPS 2020.

-----

After some careful investigation, we realized that the "major technical flaw" that *a good model for pairwise classification does not necessarily mean a good model for pointwise classification in worst cases* is actually a common issue of the entire similarity learning. Unsupervised/self-supervised representation learning, including the most promising contrastive learning such as SimCLR, also has the same issue. Thus, **this issue denies the entire similarity learning, and solving this issue is clearly beyond the scope of the current submission**.

---

> ### Comment · Area_Chair1 · 2020-11-19
> **Re: On theoretical connection from pairwise classification to pointwise classification**
>
> Thanks for the references. The bounds in the first reference are interesting, although I gather the reviewers' specific concern is the multi-class case with k >> 2.
>
> The pointer to similarity learning is also interesting. I didn't understand the reviewers' concern to be whether or not a pointwise -> pairwise reduction is sensible. Rather, I took it to specifically be whether this reduction makes the problem of learning under label noise easier. The current paper establishes that the noise on the pairwise task is lower than the pointwise one (Theorem 2). I think the question is whether the noise-reduction from working with the pairwise classifier is erased when one uses it to construct a pointwise one.
>
> R2, R5, is that a fair summary of your question on this point?

---

> > ### Author Response · Authors · 2020-11-19
> > **Re: Re: On theoretical connection from pairwise classification to pointwise classification**
> >
> > R5 explicitly said *"a lower matrix-wise error does not necessarily mean a better classification model. For a balanced large set with sufficient number of classes, a similarity model that only predicts 0(dissimilar) can achieve a risk near to 0. On the contrary, its classification risk will be large. The intuition after this, as I have stressed, is the difference between positive and negative pairs."*
> >
> > So how to parse these sentences?

---

> > ### Author Response · Authors · 2020-11-19
> > **Clarification?**
> >
> > Sorry. We discussed with each other but we cannot understand *"the noise-reduction from working with the pairwise classifier is erased when one uses it to construct a pointwise one"*. Could you please be more specific?
> >
> > Note that Theorem 3 is about the **standard** pairwise generalization error when training with **noisy** pairwise labels. In the bound, $R(\cdot)$ and $R_n(\cdot)$ are the expected\empirical pairwise risk over the **clean**-pairwise-label joint distribution which is derived from the underlying (**clean**-class-label) joint distribution, while $\hat{f}$ is the empirical pointwise model trained with **noisy** pairwise labels. We at least has proved that the standard pairwise generalization gap (the clean expected risk minus the clean empirical risk) is uniformly bounded, which holds for every pointwise model $f$ in the implicitly defined function class (a class of **size-independent** and **norm-bounded** DNNs) and thus holds for the specific $\hat{f}$.
> >
> > On the other hand, empirically speaking, Class2Simi is a **meta** method that can be applied on top of many **binary** label-noise learning methods (e.g., Forward, Reweight, and Revision; let us call it **Base**). Besides the noise reduction, we use Base to **correct** the pairwise label noise and **debias** the pairwise model being trained. As long as Base is powerful for handling noisy class labels, Base + Class2Simi will be powerful for handling noisy pairwise labels, and the clean classifier will be good after removing the additional layer in the end of training. We cannot get **why the pairwise-label-noise correction and model debiasing are erased just by removing the additional layer in the end of training**.
> >
> > Thanks!

---

> > > ### Comment · Area_Chair1 · 2020-11-19
> > > **Re: Clarification?**
> > >
> > > Thanks. To be clear, I am simply trying to condense the reviewers' concerns as I understand them, and see if we can reach some consensus.
> > >
> > > Theorem 3 makes sense: you have a generalization bound on the pairwise-learning problem. So, the empirical pairwise-risk minimizer $\hat{f}$, which maps from $X$ to $\mathbb{R}^C$, will have small clean pairwise-label error. I don't think there are concerns about this by itself.
> > >
> > > I think the question is what one can say about the clean pointwise-label error. As I understand, once you train $\hat{f}$ with your pairwise objective, you use it to make pointwise label predictions in the usual way, i.e., take $\operatorname{argmax}_i \hat{f}_i(x)$. Reviewers seem to be asking: what can we say about how well these pointwise predictions perform, either in absolute terms, or compared to the result of standard pointwise training on the noisy data? Specifically, I think the question is whether the noise-reduction observed in Theorem 2 translates to better pointwise-label error of your learned $\hat{f}$.
> > >
> > > Here is an attempt to walk through the question:
> > >
> > > (1) suppose we have a highly noisy problem. I gather this means the $T_c$ matrix you refer to is close to uniform. One can train an existing method on a sample from this problem, and get some predictor $\hat{g}$. As the noise rate is high, this will likely perform poorly on the pointwise label error.
> > >
> > > (2) suppose we apply the pairwise-learning reduction. Theorem 2 shows that the $T_s$ matrix, which is $2 \times 2$, has lower noise rate than $T_c$.
> > >
> > > (3) we now train using Class2Simi on the noisy data, and get the predictor $\hat{f}$. Owing to Theorem 3, $\hat{f}$ will generalize well on the clean data, with respect to pairwise labels. Reduction in the noise in $T_s$ corresponds to a faster rate of convergence of the empirical pairwise risk, which makes sense.
> > >
> > > (4) finally, we may compare both $\hat{g}$ and $\hat{f}$ in terms of their pointwise label error. Can $\hat{f}$ be argued to perform better than $\hat{g}$? The remarks after Theorem 2 hint that since the noise-level in $T_s$ is lower than $T_c$, $\hat{f}$ may perform better. However, the concern appears to be how one can go from a guarantee on clean pairwise error to clean pointwise error. In an extreme case of uniform $T_s$, one might learn a model with low pairwise-similarity error, but this does not translate to low pointwise-label error.
> > >
> > > **Authors**: Do the above points make sense?
> > >
> > > **R2, R5**:  Does this capture your core concern? If not, can you clarify?
> > >
> > > Your comments about Class2Simi being a meta-method are interesting. I see that the technique adds a layer on top of standard noise correction approaches, each of which produces an $\hat{f}$ which performs well on the clean pointwise-label error. In my understanding, the reviewers' question is whether this layer can strongly help: does the reduced noise in similarities translate into better learning for the pointwise error?

---

> > ### Author Response · Authors · 2020-11-19
> > **Re: Re: Clarification?**
> >
> > Thanks very much, AC, for your detailed clarification! Now we understand what the concern is. In fact, we have addressed this concern in a post entitled "on noise rate for dissimilarity labels". We cannot get the point because we thought that concern has already been addressed... Anyway, we copy and paste the post here for easy reference (it exceeds the character limit so we put it in a separate post under this one).

---

> > > ### Author Response · Authors · 2020-11-19
> > > **On noise rate for dissimilarity labels**
> > >
> > > OK, let us temporarily admit that the noise rate of negative pairs **can be very low** given fully random labels. Assume also that the noise rate of positive pairs **cannot be very high**, in order to keep the overall noise rate sufficiently low (we will overturn this assumption later). Then, this is an issue of **classification calibration**.
> > >
> > > We distinguish two scenarios: in the **practical case**, the causality is **class label-->pair label**; in the **theoretical case**, the causality is **pair label-->class label**. Considering the theoretical case, if a surrogate loss is classification-calibrated, minimizing it will leads to minimizing the zero-one loss on the class-label random variable in the limit case, and let us call it **learnable** with a bit abuse of terminology. If a loss is not classification-calibrated, we cannot guarantee the worse-case learnability, which **cannot imply average-case non-learnability either**.
> > >
> > > Note that this topic studies the limit behaviors of surrogate losses given uncountably infinite data, and the theoretical results are called **excess risk bounds** involving much more complicated theory than **estimation/generalization error bounds**. There is no guarantee that a classification-calibrated loss working well in the limit case must also work well in the finite-sample case. The latter needs estimation/generalization error bounds. What we proved is an estimation/generalization error bound guaranteeing the **weak consistency of training**, and often this is already the best we can offer in terms of the worst-case error analysis in weakly supervised learning. To the best of our knowledge, previous label-noise papers also provide such bounds rather than excess risk bounds, and very few papers studied the classification calibration (there was 1 ICML 2020 paper on binary classification under label noise).
> > >
> > > It seems fine to use non-classification-calibrated losses in practice. According to [1], the multi-class margin loss (i.e., 1-vs-rest loss) and the pairwise comparison loss (i.e., 1-vs-1 loss) are **proved to be non-calibrated**, but they are still the main multi-class losses in [2,3]. For semi-supervised learning and adversarial learning, it is even proved that **all margin-based or all convex losses are non-calibrated** [4,5]. However, people still prefer using margin-based and/or convex losses for training models in semi-supervised learning and adversarial learning. The classification calibration is just one theoretical property and not the most important one. The genuine desideratum is **a good loss that works in the practical case (finite-sample, class-->pair causality, and average-case) rather than the theoretical case (asymptotic, pair-->class causality, and worst-case)**.
> > >
> > > BTW, a key difference is that the above losses are proved to be non-calibrated while our loss is not proved to be calibrated. Hope the last paragraph will not mislead the reviewers.
> > >
> > > Last but not least, let us focus on the extreme case and look at the assumption. For class-balanced data sets with 10 classes, when the labels are random, **the noise rate of positive pairs is 0.9 and the noise rate of negative pairs is 0.1**. Pairwise learning is a special case of binary classification, and a noise rate of 0.9 is too high to learn. Even if we don't consider the noise rate of positive pairs because they are the minority class, in label-noise learning a noise rate of 0.1 is already noticeable and should be handled by the algorithm.
> > >
> > > In fact, an incorrect positive label is much more adversarial even though they are the minority class, quoted from our reply: *"In practice, it is common that one class have more than one clusters, while it is rare that two or more classes are in the same cluster. If there is a flip as similar-->dissimilar and based on it we split a (latent) cluster into two (latent) clusters, we still have some high chance to label these two clusters correctly later. If there is a flip dissimilar-->similar and based on it we join two clusters belonging to two classes into a single cluster, we nearly have zero change to label this cluster correctly later. As a consequence, the latter form is more adversarial, and without any special design of corrections, the standard pairwise training is hopeless to handle this form of label noise."*
> > >
> > > [1] Ambuj Tewari and Peter L. Bartlett, On the Consistency of Multiclass Classification Methods, JMLR 2007.
> > >
> > > [2] Mehryar Mohri, Afshin Rostamizadeh, Ameet Talwalkar. Foundations of Machine Learning, The MIT Press 2018.
> > >
> > > [3] Shai Ben-David and Shai Shalev-Shwartz. Understanding Machine Learning: From Theory to Algorithms, Cambridge University Press 2014.
> > >
> > > [4] Jesse H. Krijthe, Marco Loog. The Pessimistic Limits and Possibilities of Margin-based Losses in Semi-supervised Learning, NeurIPS 2018.
> > >
> > > [5] Han Bao, Clayton Scott, Masashi Sugiyama. Calibrated Surrogate Losses for Adversarially Robust Classification, COLT 2020.

---

> > > ### Comment · Area_Chair1 · 2020-11-19
> > > **Re: Clarification?**
> > >
> > > Great, thanks for the detailed response!
> > >
> > > **R2, R5**: could you let me know if the previous message accurately summarizes your concern? And, whether the authors' response below addresses it? Thanks.

---

> > ### Author Response · Authors · 2020-11-20
> > **More comments on the noise rate**
> >
> > In learning under class-conditional noise, the task difficulty of learning class $i$ is often defined by its **intact-vs-flipped margin** as $T_{i,i}-\max_{j\neq i}T_{j,i}$ (which is the come-from version but not the go-to version). Consider the symmetric noise and fix the noise rate. Since there are $k-1$ classes to share the noise rate, the impact of the label noise is less worse when $k$ is larger, which means the gap between learning with noisy and clean class labels should be smaller. Why the performance looks worse is because multi-class classification itself becomes more difficult when $k$ is larger (even if $n/k$ is fixed, let alone $n$ is fixed).
> >
> > Let us consider Class2Simi. When $k$ is larger, the noise rate of positive pairs is higher and the noise rate of negative pairs is lower, while the overall noise rate is lower. This partially reflects that the impact of the label noise is less worse. It cannot always reflect the impact, because the additional layer + the pairwise loss is not proved to be classification-calibrated, and sometimes we may lose the relationship between clustered classes and the original classes. We **pretrain** the multi-class classifier for 20 epochs using **noisy class labels** and hope the classifier can capture the relationship before applying Class2Simi. This is sensible since $T$ is diagonally dominant.
> >
> > The extreme case is a completely different story! Fully random class labels violate that $T$ is diagonally dominant as the intact-vs-flipped margin is 0. Now consider Class2Simi. The positive noise rate is $1-1/k$ and the negative noise rate is $1/k$. Assuming that $n\gg k$, the ratio of positive pairs is $(n-k)/(k(n-1))\approx1/k$ and the ratio of negative pairs is approximately $1-1/k$. It looks that the overall noise rate is approximately $2/k$ assuming $k\gg1$, and Class2Simi should be successful in learning with noisy pairwise labels. However, **do not forget that $T_s$ for Class2Simi should also be diagonally dominant**. The positive intact-vs-flipped margin is $1/k-(1-1/k)\approx-1$ and it is impossible for Class2Simi to learn. The original margin is 0 and it is simply **uninformative**; after the point-to-pair reduction, the margin becomes close to -1 and it is totally **adversarial**.
> >
> > Thus this extreme case cannot serve as a counterexample. A counterexample must satisfy (1) the original multi-class problem is not learnable, i.e., $T$ is not diagonally dominant; (2) the reduced binary problem is learnable, i.e., $T_s$ is diagonally dominant. The overall pairwise noise rate does not matter too much. We can see that Theorem 3 is meaningful provided that $T_{s,11}>T_{s,01}$ and $T_{s,11}$ is not too small. Thus the learnability is guaranteed by the fact that $T_s$ is diagonally dominant and after that the convergence rate is partially controlled by $1-T_{s,01}/T_{s,11}$.

---

> > ### Comment · Area_Chair1 · 2020-11-20
> > **Gentle reminder: discussion on noise rate**
> >
> > **R2, R5**: could you please take a look at the discussion below and let me know your thoughts? Thanks!
> >
> > **Authors**: a few follow-up questions, for my own understanding.
> >
> > (1) in Theorem 2, is it possible to simplify some terms? Eg, the denominators for the last two terms,
> >
> > $\sum_{i = i'} T_{c, i, j} T_{c, i', j'} = \sum_{i, j, j'} T_{c, i, j} T_{c, i, j'} = \sum_{i} (\sum_j T_{c, i, j}) (\sum_{j'} T_{c, i, j'}) = C$, with the last equality because, as I understand, the matrix $T$ has rows that sum to $1$. It looked like one could do similar simplifications for the numerator of the first two terms to $C^2 - C$.
> >
> > If this is so, the simplified forms might be a little easier to parse.
> >
> > (2) if I understand, your argument about the inapplicability of the the extreme case of uniform $T_c$ is that the resulting $T_s$ will have negative intact-vs-flipped margin, and so learning on pairs won't be possible. This makes sense to me.
> >
> > Just to summarize in one place, could you comment about whether it's true in general that:
> >
> > (2a) if $T_c$ has strictly positive intact-vs-flipped margin, then $T_s$ will also. Eg, I am curious what happens if $T_c$ has diagonal elements $\frac{1}{C} + \epsilon$, and off-diagonal elements $\frac{1}{C} - \frac{\epsilon}{C - 1}$.
> >
> > (2b) if $T_s$ has strictly positive intact-vs-flipped margin, then $T_c$ will also. As I understand this would rule out a learnability counter-example, so would be of interest.
> >
> > Thanks!

---

> > > ### Author Response · Authors · 2020-11-21
> > > **Re: "a few follow-up questions, for my own understanding" (part 1)**
> > >
> > > Thanks very much for the thought-provoking questions! We first address part 1.
> > >
> > > We are very sorry that we didn't simplify them. In fact,
> > >
> > > $T_{s,00} = 1-[\sum_j(\sum_iT_{c,ij})^2-||T_c||_{\mathrm{Fro}}^2]/(C^2-C)$,
> > >
> > > $T_{s,11} = ||T_c||_{\mathrm{Fro}}^2/C$.
> > >
> > > If $T_c$ is symmetric noise or pairflip noise (or any CCN that is also column-normalized), we also have $\sum_iT_{c,ij}=1$, and
> > >
> > > $T_{s,00} = 1-(C-||T_c||_{\mathrm{Fro}}^2)/(C^2-C)$.

---

> > > ### Author Response · Authors · 2020-11-21
> > > **Re: "a few follow-up questions, for my own understanding" (part 2a)**
> > >
> > > We are very sorry again that we made a mistake in a previous post. Binary classification is quite special, and it is learnable as long as $T_{s,01}+T_{s,10}<1$ or $T_{s,00}+T_{s,11}>1$ (see for example "Learning from Corrupted Binary Labels via Class-Probability Estimation", ICML 2015). $T_{s,00}+T_{s,11}>1$ implies $T_s$ is diagonally dominant in one direction and the intact-vs-flipped margin is strictly positive in one version (among go-to and come-from versions), so $T_s$ does not need to be diagonally dominant in both directions and the intact-vs-flipped margin does not need to be strictly positive in both versions... Of course, if these two conditions can be satisfied, then noise correction and model debiasing would be much easier.
> > >
> > > In our problem of interest, $T_{s,11}>T_{s,01}$ is much more possible than $T_{s,11}>T_{s,10}$, i.e., $T_s$ is likely to be diagonally dominant in the column direction and the come-from version of the intact-vs-flip margin is likely to be positive (which is consistent with Theorem 3).
> > >
> > > Consider the symmetric noise (or any CCN that is also column-normalized). It holds that
> > >
> > > $T_{s,00}+T_{s,11}-1=(||T_c||_{\mathrm{Fro}}^2-1)/(C-1)$,
> > >
> > > so **it is learnable if and only if $||T_c||_{\mathrm{Fro}}>1$**.
> > >
> > > When $T_c$ is fully uniform, $||T_c||_{\mathrm{Fro}}=1$; when it slightly deviates from being fully uniform, $||T_c||_{\mathrm{Fro}}>1$ can be guaranteed. Hence, given the symmetric noise, the learnability of the original multi-class classification always implies the learnability of the reduced binary classification.
> > >
> > > Note that within CCN, the symmetric noise is least adversarial and thus allows the largest noise rate. As a consequence, **given any column-normalized CCN, the learnability of the original multi-class classification always implies the learnability of the reduced binary classification**.
> > >
> > > -----
> > >
> > > We would like to clarify that the go-to intact-vs-flip margin $T_{i,i}-\max_{j\neq i}T_{i,j}$ is always meaningful, while the come-from version $T_{i,i}-\max_{j\neq i}T_{j,i}$ is only meaningful if classes are balanced. Binary classification is too special that both versions are meaningful no matter if two classes are balanced or not. If the come-from version is positive for two classes, the go-to version is positive at least for one of the two classes. After we successfully noise-correct one class, the other class is done as there are just two classes.

---

> > > ### Author Response · Authors · 2020-11-21
> > > **Re: "a few follow-up questions, for my own understanding" (part 2b)**
> > >
> > > Unfortunately, (2b) is not true. A set of sufficient but not necessary conditions of (2b) may include (A) ERM-based similarity learning is (multi-class) classification-calibrated and (B) the original $T_c$ is not too adversarial. However, (A) is still the biggest open problem in similarity learning. We address (A) by pointwise pretraining on noisy class labels, which works well in practice but cannot be reflected in theory.
> > >
> > > Consider the most adversarial CCN, i.e., the pairflip noise. So long as $T_{c,ii}\neq1/2$, we must have $T_{s,11}>1/2$ which is very strong. This means the reduced similarity learning is an easy problem, of course, **from the similarity learning or clustering point of view**. In an extreme case, let $r_1,...,r_C$ be a permutation of $1,...,C$ and let $T_{c,i,r_i}=1$. Then the original noise rate is 1 but the reduced noise rate is 0. In other words, we can perform clustering by similarity learning but we cannot perform classification itself, since we have already **lost the relationship between clusters and classes** due to the permutation. In practice, our pretraining can help to recover this relationship whenever $T_{c,ii}>1/2$.
> > >
> > > Even if we consider the least adversarial CCN, i.e., the symmetric noise, (2b) may not be true. As we replied, when $T_c$ slightly deviates from being fully uniform, the learnability of similarity learning can be guaranteed, **so this actually covers $-1/C\le\epsilon<0$**. When $-1/C\le\epsilon<0$, the original multi-class classification is no longer learnable in the sense of learning with noisy labels (all sample selection/reweighting and label correction will fail), but it is still learnable in the sense of learning with complementary labels (loss correction is still possible). For such negative $\epsilon$, we can perform similarity learning but multi-class classification by Class2Simi must also fail because **our pretraining becomes misleading**, similarly to the pairflip noise when $T_{c,ii}<1/2$.

---

> > > ### Comment · Area_Chair1 · 2020-11-21
> > > **Re: a few follow-up questions**
> > >
> > > Thanks for the responses!
> > >
> > > For question (1), thanks for the confirmation. I'd suggest using the simplified versions, as they make the expressions clearer.
> > >
> > > For question (2), is this a fair summary? Learnability of the binary similarity problem is:
> > >
> > > (P1) in general, equivalent to $T_{s, 01} + T_{s, 10} < 1$. This doesn't require positive "go-to" intact-vs-flip margin for both classes.
> > >
> > > (P2) for column-normalized $T_c$, implied by (in fact equivalent to) the original classification problem satisfying $|| T_{c} ||_{{\rm F}} > 1$.
> > >
> > > (P3) in general, not sufficient to imply learnability of the original classification problem. Eg, if one deterministically permutes the labels. The pre-training step aims to reduce this effect.
> > >
> > > ---
> > >
> > > As I find this discussion to be useful, I have a couple of further follow-up clarifications:
> > >
> > > (Q1) suppose the noise in $T_c$ is **not** column-normalized. Here could one have a $T_c$ that is learnable, but a $T_s$ that is not? Here, let "learnable" be the most general sense, including the "complementary label" setting you mention. Maybe your pairwise flip example discusses such a setting, but just want to check.
> > >
> > > (Q2) suppose the original noisy classification problem is **not** learnable, even in the "complementary label" setting. Eg, say $T_c$ is non-invertible. Here could one get a learnable $T_s$?
> > >
> > > Eg, consider $T_c = \begin{bmatrix} I_{C/2} & 0_{C/2} \\\\
> > > I_{C/2} & 0_{C/2} \end{bmatrix}$, where ${I}$ is the identity and ${0}$ the zeros matrix, and $C$ assumed to be even. Would we get $T_{s, 11} = 1$ and $T_{s, 00} > 0$?
> > >
> > > If I've misunderstood anything, please let me know. Thanks!

---

> > > ### Comment · Area_Chair1 · 2020-11-22
> > > **Regarding generalization bounds vs classification calibration**
> > >
> > > **Authors**: I'd like to request a clarification on the discussion about generalization bounds vs classification calibration.
> > >
> > > Let $\hat{f}$ denote the minimizer of the empirical pairwise surrogate loss. As I understand, there are four distinct possible bounds:
> > >
> > > (1) a generalization error bound on the **pairwise** labels. Ie, what is the pairwise surrogate risk of $\hat{f}$ on a test sample?
> > >
> > > (2) a surrogate regret bound on the **pairwise** labels. Ie, what does a low pairwise surrogate risk of $\hat{f}$ tell us about the pairwise 0-1 risk?
> > >
> > > (3) a generalization error bound on the **pointwise** labels. Ie, what is the pointwise surrogate risk of $\hat{f}$ on a test sample?
> > >
> > > (4) a surrogate regret bound on the **pointwise** labels. Ie, what does a low pairwise surrogate risk of $\hat{f}$ tell us about the pointwise 0-1 risk?
> > >
> > > My understanding is:
> > >
> > > (1) is provided by Theorem 3.
> > >
> > > (2) is a consequence of existing surrogate regret bounds for binary problems.
> > >
> > > (4) is provided by the paper "Similarity-based Classification: Connecting Similarity Learning to Binary Classification" for the case $C = 2$. The case $C > 2$ is unknown.
> > >
> > > Is this accurate? Could you also clarify whether there is a bound for (3)? Thanks.

---

> > > ### Author Response · Authors · 2020-11-23
> > > **Two learnability types**
> > >
> > > Thanks for the (almost) perfect summary of the learnability and sorry for the late reply. Please allow me to further distinguish two cases, with/without knowing $T$.
> > >
> > > CCN can be rewritten as $\mathbf{\bar{p}}=T\mathbf{p}$, where $\mathbf{p}$ and $\mathbf{\bar{p}}$ are vectors holding the clean and corrupted class-posterior probabilities. When talking about the learnability, $\mathbf{\bar{p}}$ can be regarded as given (in addition to $p(x)$). Subsequently, if $T$ is known, the learnability is same as the **invertibility of $T$**. This applies to $T_c$ and $T_s$.
> > >
> > > On the other hand, if $T$ is unknown, we need also to consider the **identifiability of $T$** based on $\mathbf{\bar{p}}$. To this end, people often assume that **anchor points exist**, in particular for image classification, people always assume it. An anchor point is some $x$ where $p(Y=i|X=x)=1$ for some class $i$. A single anchor point $x$ for class $i$ allows us to recover the whole $i$-th column of $T$ as $\mathbf{\bar{p}}(x)$. However, given some anchor points, how can we know **which anchor point is for which class**? Here comes the play of being diagonally dominant: if an anchor point is classified into the noisy class $i$, it must be for the clean class $i$. This was the learnability in our discussion in order to identify $T$ from $\mathbf{\bar{p}}$ or even estimate $T$ from only corrupted data.

---

> > > ### Author Response · Authors · 2020-11-23
> > > **Proof about the learnability**
> > >
> > > Now let us prove the original learnability implies the reduced learnability in the most general sense, i.e., $T_c$ is invertible.
> > >
> > > Denote by $\mathbf{v}_j$ the $j$-th column of $T_c$ and $\mathbf{1}$ the all-one vector. Then,
> > >
> > > $\sum_j(\sum_iT_{c,ij})^2
> > > = \sum_j\langle\mathbf{v}_j,\mathbf{1}\rangle^2
> > > \le \sum_j||\mathbf{v}_j||^2||\mathbf{1}||^2
> > > = C||T_c||_{\mathrm{Fro}}^2$
> > >
> > > where we used the Cauchy–Schwarz inequality in the second step.
> > >
> > > The above result leads to $T_{s,11}+T_{s,00}\ge1$. Note that the equality holds only if all $\mathbf{v}_j$ aligns with $\mathbf{1}$ directionally, which means $T_c$ is fully uniform and thus non-invertible. This is what we were to prove.
> > >
> > > This argument shows the original learnability with knowing $T_c$ can imply the reduced learnability with knowing neither $T_c$ nor $T_s$. In other words, the reduced label-noise learning is indeed **easier** than the original label-noise learning.

---

> > > ### Author Response · Authors · 2020-11-23
> > > **Regarding non-invertible $T_c$**
> > >
> > > Yes, this is a counterexample when $T_c$ is known, where the original problem is non-learnable and the reduced problem is learnable.
> > >
> > > Note that, however, the proposal has **two components**: pointwise pretraining and pairwise training. The latter being learnable alone cannot guarantee the combination of them to be learnable... The pretraining needs to **roughly** capture the relationship between noisy and clean classes, and its learnability is **harder** requiring $T_c$ to be clearly diagonally dominant.
> > >
> > > After assuming $T_c$ is known, we may modify the pretraining to **more advanced pretraining** such as **backward/forward loss correction**. Then the learnability of pretraining is exactly same as the original learnability. We have already shown that the pairwise training is easier to guarantee its learnability. As a consequence, if $T_c$ is known, the learnability of **improved** pointwise pretraining + pairwise training should coincide with the original learnability. Just the effect of pretraining is very hard to analyze in theory.
> > >
> > > Whenever $T_c$ is non-invertible, the transformation from $\mathbf{p}$ to $\mathbf{\bar{p}}$ is **a lossy channel**, and we cannot expect the pretraining to be lossless. In this example, two classes are relabeled as one class $C/2$ times and there is no other corruption. The pretraining will do the same thing, i.e., mapping two classes to one class, and this cannot be mitigated since $T_c$ is non-invertible. In this sense, this counterexample considering the pairwise training is **not a counterexample anymore** considering the pointwise pretraining + pairwise training.
> > >
> > > PS, if $T_c$ is not given for training, similar pretraining is also indispensable to learning $\mathbf{\bar{p}}$, finding anchor points, and estimating $T_c$. Therefore, our pretraining is **not additional** to previous loss correction or label correction methods where $T_c$ is used but not given.

---

> > > ### Comment · Area_Chair1 · 2020-11-23
> > > **Re: answers to (Q1) and (Q2)**
> > >
> > > Thanks for the responses, which are helpful. I think the paper would benefit from some discussion of a summarized version of these points, as they are not immediately apparent.

---

> > > ### Author Response · Authors · 2020-11-23
> > > **Re: Regarding generalization bounds vs classification calibration**
> > >
> > > We would like to take a closer look at (G) generalization error bounds $R(f)-\widehat{R}(f)$ (where $\sup_{f\in\mathcal{F}}$ is omitted for simplicity) vs (E) estimation error bounds $R(\hat{f})-R(f^*)$. Both of them are proved by the same technique, namely **uniform deviation bounds**. However, (G) holds for all $f$ without involving training, and (E) holds only for $\hat{f}$ where training is already finished.
> > >
> > > They represent **different philosophies**. (G) reasons as follows: with high probability, any $f$ can generalize well; this is guaranteed before training and it doesn't matter which $f$ is selected after training. While (G) is quite modern (the first paper was in 1974 in Russian, its German translation was in 1986, and its English translation may be after 1990), (E) originates from the estimation theory in statistics (looking like $||\hat{\theta}-\theta^*||^2\le\cdots$) and is what people considered for a few hundred years (can date at least back to Sir Isaac Newton). By **generalize well**, we meant an expected performance measure is only a bit worse than the corresponding empirical version. As this expected performance measure $R(\hat{f})$ is of our interest, the **reference** is thus the empirical performance measure $\widehat{R}(\hat{f})$. Note that **the loss in (G) is not necessarily the one for training** since training is not really involved: if it is the same loss, $\widehat{R}(\hat{f})$ will be directly minimized and the bound is very strong; otherwise, $\widehat{R}(\hat{f})$ may not be so small and the bound may not be strong. On the other hand, **the loss in (E) must be the one for training**, otherwise we cannot obtain (E) from the underlying uniform deviation bound. As a result, the reference for $R(\hat{f})$ is $R(f^*)$, which is by definition the **strongest reference** within $\mathcal{F}$.
> > >
> > > Now we can address (3) a generalization error bound on the pointwise labels. Since we can prove (1) a generalization error bound on the pairwise labels, we can similarly prove (3) by changing the loss under consideration. We can also prove (1') an estimation error bound on the pairwise labels, because the loss in (1) is exactly the one for training. What we cannot prove is (3') an estimation error bound or excess risk bound on the pointwise labels where the excess risk is the sum of the estimation and approximation errors. If a pairwise loss is calibrated, we will have
> > >
> > > $R_c(\hat{f})-\inf R_c(f)\le\psi(R_s(\hat{f})-\inf R_s(f))$
> > >
> > > where $R_c$ is the pointwise risk, $R_s$ is the pairwise risk, the infima are taken over all measurable functions instead of over $\mathcal{F}$, and $\psi$ is a convex, invertible, non-decreasing function passing through the origin (i.e, $\psi(0)=0$). This is too difficult to prove and should go beyond the scope of the current submission.
> > >
> > > If necessary, we can provide (3), though it is not very meaningful. When training is involved, $\widehat{R}_s(f)$ is minimized but not $\widehat{R}_c(f)$, and (3) can be theoretically bounded just because a weak reference is chosen.

---

> > > ### Comment · Area_Chair1 · 2020-11-23
> > > **Re: Regarding generalization bounds vs classification calibration**
> > >
> > > Thanks, this clarifies about the four different bounds.
> > >
> > > To tie back to the start of this thread, could I request that you briefly summarize what we can theoretically argue about the **pointwise** classification performance of the Class2Simi **pairwise** reduction? I realize this has been gone over at length, but it would be useful to put everything together concisely.

---

> > > > ### Author Response · Authors · 2020-11-24
> > > > **Summary of the discussions on theory**
> > > >
> > > > We are interested in three problem settings:
> > > >
> > > > (**P1**) Learning with pointwise class labels with label noise. The aim is to remove the side-effect of label noise.
> > > >
> > > > (**P2a**) Pre-train the model by using the pointwise class labels with label noise. The aim is to match the identity of the $i$-th clean class to the identity of the $i$-th noisy class. We do not need to reduce label noise in this problem.
> > > >
> > > > (**P2b**) Learning with pairwise similarity labels with label noise. The aim is to remove the side-effect of label noise.
> > > > The conclusions are as follows:
> > > >
> > > > (1) If **P1** is learnable, so is **P2b**, but not vise versa. A counterexample is provided in the discussion.
> > > >
> > > > (2) Although **P1** and **P2a** have different aims, they have the same requirement for learnability.
> > > >
> > > > (3) If the noise transition matrix **$T_c$** is not available, all the three settings require the transition matrix to be diagonally dominant to guarantee learnability (we do not require $T_s$ to be diagonally dominant; note that $T_c$ implies $T_s$, but not vise versa). If the transition matrix **$T_c$** is given, the three settings are learnable if the transition matrix is invertible.
> > > >
> > > > (4) The learnability of Class2Simi (or **P2**) is decided by the learnability of **P2a** because **P2b** is not learnable only when the similarity labels are completely random.

---

### Author Response · Authors · 2020-11-24
**Revised submission uploaded**

Dear reviewers and all,

We have revised our draft according to all the valuable comments. Major revisions are highlighted in green. We sincerely thank all the reviewers. We would highly appreciate it if you could read our responses and revisions. Please feel free to let us know if further details/explanations would be helpful.

Best,

Authors

---

### Decision · Program_Chairs · 2021-01-07
**Final Decision**

**Decision:**

Reject

**Comment:**

The paper's stated contributions are:

(1) a new perspective on learning with label noise, which reduces the problem to a similarity learning (Ie, pairwise classification) task

(2) a technique leveraging the above to learn from noisy similarity labels, and a theoretical analysis of the same

(3) empirical demonstration that the proposed technique surpasses baselines on real-world benchmarks

Reviewers agreed that (1) is an interesting new perspective that is worthy of study. In the initial set of reviews, there were concerns about (2) and (3); for example, there were questions on whether the theoretical analysis studies the "right" quantity (pointwise vs pairwise loss), and a number of questions on the experimental setup and results (Eg, the computational complexity of the technique). Following a lengthy discussion, the authors clarified some of these points, and updated the paper accordingly.

At the conclusion of the discussion, three reviewers continued to express concerns on the following points:

- *Theoretical justification*. Following Theorem 3, the authors assert that their results "theoretically justifies why the proposed method works well". The analysis indeed provides some interesting properties of the reduction, such as the fact that it preserves learnability (Appendix F), and that the "total noise" is reduced (Theorem 2). However, a complete theoretical justification would involve guaranteeing that the quantity of interest (Ie, the clean pointwise classification risk) is guaranteed to be small under the proposed technique. Such a guarantee is lacking.
  - This is not to suggest that such a guarantee is easy -- as the authors note, this might involve a bound that relates pointwise and pairwise classification in multi-class settings, and such bounds have only recently been shown for binary problems -- or necessary for their method being practical useful (per discussion following Theorem 3). Nonetheless, without such a bound, there are limits to what the current theory justifies about the technique's performance in terms of the final metric of interest.

- *Comparison to SOTA*. Reviewers noted that the gains of the proposed technique are often modest, with the exception of CIFAR-100 with high noise. Further, the best performing results are significantly worse than those reported in two recent works, namely, Iterative-CV and DivideMix. The authors responded to the former in the discussion, and suggested that they might be able to combine results with the latter. While plausible, given that the latter sees significant gains (Eg, >40% on CIFAR-100), concrete demonstration of this point is advisable: it is not immediately apparent to what extent the gains of the proposed technique seen on "simple" methods (Eg, Forward) would translate more "complex" ones (Eg, DivideMix).
  - In the response, the authors also mentioned that (at least the initial batch of) the experiments are intended to be a proof-of-concept. This would be perfectly acceptable for a work with a strong theoretical justification. However, per above, this point is not definitive.

- *Creation of Clothing1M*. The authors construct a variant of Clothing1M which merges the classes 3 and 5. Given that prior work compares methods on the original data, and that this potentially reflects noise one may encounter in some settings, it is advisable to at least report results on the original, unmodified version.

- *Issues with clarity*. There are some grammatical issues (Eg, "is exact the"), typos (Eg, "over 3 trails"), notational inconsistencies (Eg, use of C for # of classes in Sec 2, but then c in Sec 3.1), and imprecision in explanation (Eg, Sec 3.2 could be clearer what precise relationships are used from [Hsu et al. 2019]).
  - These are minor but ought to be fixed with a careful proof-read.

Cumulatively, these points suggest that the work would be served by further revision and review. The authors are encouraged to incorporate the reviewers' detailed comments.